# META-LEARNING WITH NEGATIVE LEARNING RATES

**Alberto Bernacchia**
MediaTek Research
`alberto.bernacchia@mtkresearch.com`

## ABSTRACT

Deep learning models require a large amount of data to perform well. When data is scarce for a target task, we can transfer the knowledge gained by training on similar tasks to quickly learn the target. A successful approach is *meta-learning*, or *learning to learn* a distribution of tasks, where *learning* is represented by an outer loop, and *to learn* by an inner loop of gradient descent. However, a number of recent empirical studies argue that the inner loop is unnecessary and more simple models work equally well or even better. We study the performance of MAML as a function of the learning rate of the inner loop, where zero learning rate implies that there is no inner loop. Using random matrix theory and exact solutions of linear models, we calculate an algebraic expression for the test loss of MAML applied to mixed linear regression and nonlinear regression with overparameterized models. Surprisingly, while the optimal learning rate for adaptation is positive, we find that the optimal learning rate for training is always negative, a setting that has never been considered before. Therefore, not only does the performance increase by decreasing the learning rate to zero, as suggested by recent work, but it can be increased even further by decreasing the learning rate to negative values. These results help clarify under what circumstances meta-learning performs best.

## 1 INTRODUCTION

Deep Learning models represent the state-of-the-art in several machine learning benchmarks (LeCun et al. (2015)), and their performance does not seem to stop improving when adding more data and computing resources (Rosenfeld et al. (2020), Kaplan et al. (2020)). However, they require a large amount of data and compute to start with, which are often not available to practitioners. The approach of *fine-tuning* has proved very effective to address this limitation: pre-train a model on a source task, for which a large dataset is available, and use this model as the starting point for a quick additional training (fine-tuning) on the small dataset of the target task (Pan & Yang (2010), Donahue et al. (2014), Yosinski et al. (2014)). This approach is popular because pre-trained models are often made available by institutions that have the resources to train them.

In some circumstances, multiple source tasks are available, all of which have scarce data, as opposed to a single source task with abundant data. This case is addressed by *meta-learning*, in which a model gains experience over multiple source tasks and uses it to improve its learning of future target tasks. The idea of meta-learning is inspired by the ability of humans to generalize across tasks, without having to train on any single task for long time. A meta-learning problem is solved by a bi-level optimization procedure: an outer loop optimizes meta-parameters across tasks, while an inner loop optimizes parameters within each task (Hospedales et al. (2020)).

The idea of meta-learning has gained some popularity, but a few recent papers argue that a simple alternative to meta-learning is just good enough, in which the inner loop is removed entirely (Chen et al. (2020a), Tian et al. (2020), Dhillon et al. (2020), Chen et al. (2020b), Raghu et al. (2020)). Other studies find the opposite (Goldblum et al. (2020), Collins et al. (2020), Gao & Sener (2020)). It is hard to resolve the debate because there is little theory available to explain these findings.

In this work, using random matrix theory and exact solutions of linear models, we derive an algebraic expression of the average test loss of MAML, a simple and successful meta-learning algorithm (Finn et al. (2017)), as a function of its hyperparameters. In particular, we study its performance as a

function of the inner loop learning rate during meta-training. Setting this learning rate to zero is equivalent to removing the inner loop, as advocated by recent work (Chen et al. (2020a), Tian et al. (2020), Dhillon et al. (2020), Chen et al. (2020b), Raghu et al. (2020)). Surprisingly, we find that the optimal learning rate is negative, thus performance can be increased by reducing the learning rate below zero. In particular, we find the following:

- In the problem of mixed linear regression, we prove that the optimal learning rate is always negative in overparameterized models. The same result holds in underparameterized models provided that the optimal learning rate is small in absolute value. We validate the theory by running extensive experiments.

- We extend these results to the case of nonlinear regression and wide neural networks, in which the output can be approximated by a linear function of the parameters (Jacot et al. (2018), Lee et al. (2019)). While in this case we cannot prove that the optimal learning rate is always negative, preliminary experiments suggest that the result holds in this case as well.

## 2   RELATED WORK

The field of meta-learning includes a broad range of problems and solutions, see Hospedales et al. (2020) for a recent review focusing on neural networks and deep learning. In this context, meta-learning received increased attention in the past few years, several new benchmarks have been introduced, and a large number of algorithms and models have been proposed to solve them (Vinyals et al. (2017), Bertinetto et al. (2019), Triantafillou et al. (2020)). Despite the surge in empirical work, theoretical work is still lagging behind.

Similar to our work, a few other studies used random matrix theory and exact solutions to calculate the average test loss for the problem of linear regression (Advani & Saxe (2017), Hastie et al. (2019), Nakkiran (2019)). To our knowledge, our study is the first to apply this technique to the problem of meta-learning with multiple tasks. Our results reduce to those of linear regression in the case of one single task. Furthermore, we are among the first to apply the framework of Neural Tangent Kernel (Jacot et al. (2018), Lee et al. (2019)) to the problem of meta-learning (a few papers appeared after our submission: Yang & Hu (2020), Wang et al. (2020a), Zhou et al. (2021)).

Similar to us, a few theoretical studies looked at the problem of mixed linear regression in the context of meta-learning. In Denevi et al. (2018), Bai et al. (2021), a meta-parameter is used to bias the task-specific parameters through a regularization term. Kong et al. (2020) looks at whether many tasks with small data can compensate for a lack of tasks with big data. Tripuraneni et al. (2020), Du et al. (2020) study the sample complexity of representation learning. However, none of these studies look into the effect of learning rate on performance, which is our main focus.

In this work, we focus on MAML, a simple and successful meta-learning algorithm (Finn et al. (2017)). A few theoretical studies have investigated MAML, looking at: universality of the optimization algorithm (Finn & Levine (2018)), bayesian inference interpretation (Grant et al. (2018)), proof of convergence (Ji et al. (2020)), difference between convex and non-convex losses (Saunshi et al. (2020)), global optimality (Wang et al. (2020b)), effect of the inner loop (Collins et al. (2020), Gao & Sener (2020)). Again, none of these studies look at the effect of the learning rate, the main subject of our work. The theoretical work of Khodak et al. (2019) connects the learning rate to task similarity, while the work of Li et al. (2017) meta-learns the learning rate.

## 3   META-LEARNING AND MAML

In this work, we follow the notation of Hospedales et al. (2020) and we use MAML (Finn et al. (2017)) as the meta-learning algorithm. We assume the existence of a distribution of tasks $\tau$ and, for each task, a loss function $\mathcal{L}^\tau$ and a distribution of data points $\mathcal{D}^\tau = \{x^\tau, y^\tau\}$ with input $x^\tau$ and label $y^\tau$. We assume that the loss function is the same for all tasks, $\mathcal{L}^\tau = \mathcal{L}$, but each task is characterized by a different distribution of the data. The empirical meta-learning loss is evaluated on a sample of

$m$ tasks, and a sample of $n_v$ validation data points for each task:

$$\mathcal{L}^{meta}\left(\boldsymbol{\omega}; \mathcal{D}_t, \mathcal{D}_v\right) = \frac{1}{mn_v} \sum_{i=1}^{m} \sum_{j=1}^{n_v} \mathcal{L}\left(\boldsymbol{\theta}(\boldsymbol{\omega}; \mathcal{D}_t^{(i)}); x_j^{v(i)}, y_j^{v(i)}\right) \tag{1}$$

The training set $\mathcal{D}_t^{(i)} = \left\{x_j^{t(i)}, y_j^{t(i)}\right\}_{j=1:n_t}$ and validation set $\mathcal{D}_v^{(i)} = \left\{x_j^{v(i)}, y_j^{v(i)}\right\}_{j=1:n_v}$ are drawn independently from the same distribution in each task $i$. The function $\boldsymbol{\theta}$ represents the adaptation of the meta-parameter $\boldsymbol{\omega}$, which is evaluated on the training set. Different meta-learning algorithms correspond to a different choice of $\boldsymbol{\theta}$, we describe below the choice of MAML (Eq.3), the subject of this study. During *meta-training*, the loss of Eq.1 is optimized with respect to the meta-parameter $\boldsymbol{\omega}$, usually by stochastic gradient descent, starting from an initial point $\boldsymbol{\omega}_0$. The optimum is denoted as $\boldsymbol{\omega}^\star(\mathcal{D}_t, \mathcal{D}_v)$. This optimization is referred to as the *outer loop*, while computation of $\boldsymbol{\theta}$ is referred to as the *inner loop* of meta-learning. During *meta-testing*, a new (target) task is given and $\boldsymbol{\theta}$ adapts on a set $\mathcal{D}_r$ of $n_r$ target data points. The final performance of the model is computed on test data $\mathcal{D}_s$ of the target task. Therefore, the test loss is equal to

$$\mathcal{L}^{test} = \mathcal{L}^{meta}\left(\boldsymbol{\omega}^\star(\mathcal{D}_t, \mathcal{D}_v); \mathcal{D}_r, \mathcal{D}_s\right) \tag{2}$$

In MAML, the inner loop corresponds to a few steps of gradient descent, with a given learning rate $\alpha_t$. In this work we consider the simple case of a single gradient step:

$$\boldsymbol{\theta}(\boldsymbol{\omega}; \mathcal{D}_t^{(i)}) = \boldsymbol{\omega} - \frac{\alpha_t}{n_t} \sum_{j=1}^{n_t} \frac{\partial \mathcal{L}}{\partial \theta}\bigg|_{\boldsymbol{\omega}; x_j^{t(i)}, y_j^{t(i)}} \tag{3}$$

If the learning rate $\alpha_t$ is zero, then parameters are not adapted during meta-training and $\boldsymbol{\theta}(\boldsymbol{\omega}) = \boldsymbol{\omega}$. In that case, a single set of parameters in learned across all data and there is no inner loop. However, it is important to note that a distinct learning rate $\alpha_r$ is used during meta-testing. A setting similar to this has been advocated in a few recent studies (Chen et al. (2020a), Tian et al. (2020), Dhillon et al. (2020), Chen et al. (2020b), Raghu et al. (2020)).

We show that, intuitively, the optimal learning rate at meta-testing (adaptation) time $\alpha_r$ is always positive. Surprisingly, in the family of problems considered in this study, we find that the optimal learning rate during meta-training $\alpha_t$ is instead negative. We note that the setting $\alpha_t = 0$ effectively does not use the $n_t$ training data points, therefore we could in principle add this data to the validation set, but we do not consider this option here since we are interested in a wide range of possible values of $\alpha_t$ as opposed to the specific case $\alpha_t = 0$.

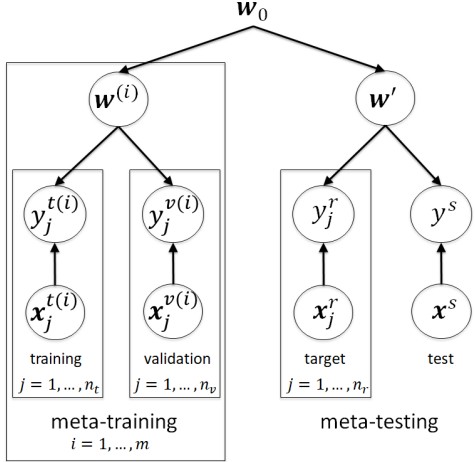

## 4 MIXED LINEAR REGRESSION

We study MAML applied to the problem of mixed linear regression. Note that the goal here is not to solve the problem of mixed linear regression, but to probe the performance of MAML as a function of its hyperparameters.

Figure 1: Graphical model of data generation in mixed linear regression

In mixed linear regression, each task is characterized by a different linear function, and a model is evaluated by the mean squared error loss function. We assume a generative model in the form of $y = \mathbf{x}^T \mathbf{w} + z$, where $\mathbf{x}$ is the input vector (of dimension $p$), $y$ is the output (scalar), $z$ is noise (scalar), and $\mathbf{w}$ is a vector of generating parameters (of dimension $p$), therefore $p$ represents both the number of parameters and the input dimension. All distributions are assumed Gaussian:

$$\mathbf{w} \sim \mathcal{N}\left(\mathbf{w}_0, \frac{\nu^2}{p} I_p\right) \qquad \mathbf{x} \sim \mathcal{N}(0, I_p) \qquad y|\mathbf{x}, \mathbf{w} \sim \mathcal{N}\left(\mathbf{x}^T \mathbf{w}, \sigma^2\right) \tag{4}$$

where $I_p$ is the $p \times p$ identity matrix, $\sigma$ is the label noise, $\mathbf{w}_0$ is the task mean and $\nu$ represents the task variability. Different meta-training tasks $i$ correspond to different draws of generating parameters $\mathbf{w}^{(i)}$, while the parameters for the meta-testing task are denoted by $\mathbf{w}'$. We denote by superscripts $t$, $v$, $r$, $s$ the training, validation, target and test data, respectively. A graphical model of data generation is shown in Figure 1.

Using random matrix theory and exact solutions of linear models, we calculate the test loss as a function of the following hyperparameters: the number of training tasks $m$, number of data points per task for training ($n_t$), validation ($n_v$) and target ($n_r$), learning rate for training $\alpha_t$ and for adaptation to target $\alpha_r$. Furthermore, we have the hyperparameters specific to the mixed linear regression problem: $p$, $\nu$, $\sigma$, $\mathbf{w}_0$. Since we use exact solutions to the linear problem, our approach is equivalent to running the outer loop optimization until convergence (see section 7.1 in the Appendix for details). We derive results in two cases: overparameterized $p > n_v m$ and underparameterized $p < n_v m$.

# 5 RESULTS

## 5.1 OVERPARAMETERIZED CASE

In the overparameterized case, the number of parameters $p$ is larger than the total number of validation data across tasks $n_v m$. In this case, since the data does not fully constrain the parameters, the optimal value of $\boldsymbol{\omega}$ found during meta-training depends on the initial condition used for optimization, which we call $\boldsymbol{\omega}_0$.

**Theorem 1.** *Consider the algorithm of section 3 (MAML one-step), and the data generating model of section 4 (mixed linear regression). Let $p > n_v m$. Let $p(\xi)$ and $n_t(\xi)$ be any function of order $O(\xi)$ as $\xi \to \infty$. Let $|\boldsymbol{\omega}_0 - \mathbf{w}_0|$ be of order $O(\xi^{-1/4})$. Then the test loss of Eq.2, averaged over the entire data distribution (see Eq.27 in the Appendix) is equal to*

$$\overline{\mathcal{L}}^{test} = \frac{\sigma^2}{2}\left(1 + \frac{\alpha_r^2 p}{n_r}\right) +$$

$$+ h^r \left[\frac{\nu^2}{2}\left(1 + \frac{n_v m}{p}\right) + \frac{1}{2}\left(1 - \frac{n_v m}{p}\right)|\boldsymbol{\omega}_0 - \mathbf{w}_0|^2 + \frac{\sigma^2 n_v m}{2p}\frac{1 + \frac{\alpha_t^2 p}{n_t}}{h^t}\right] + O\left(\xi^{-3/2}\right) \quad (5)$$

*where we define the following expressions*

$$h^t = (1 - \alpha_t)^2 + \alpha_t^2 \frac{p+1}{n_t} \quad (6)$$

$$h^r = (1 - \alpha_r)^2 + \alpha_r^2 \frac{p+1}{n_r} \quad (7)$$

*Proof.* The proof of this Theorem can be found in the Appendix, sections 7.3, 7.3.1. $\square$

The loss always increases with the output noise $\sigma$ and task variability $\nu$. Overfitting is expressed in Eq.5 by the term $|\boldsymbol{\omega}_0 - \mathbf{w}_0|$, the distance between the initial condition for the optimization of $\boldsymbol{\omega}_0$ and the ground truth mean of the generating model $\mathbf{w}_0$. Adding more validation data $n_v$ and tasks $m$ may increase or decrease the loss depending on the size of this term relative to the noise (Nakkiran (2019)), as it does reducing the number of parameters $p$. However, the loss always decreases with the number of data points for the target task $n_r$, as that data only affects the adaptation step.

Our main focus is studying how the loss is affected by the learning rates, during training $\alpha_t$ and adaptation $\alpha_r$. The loss is a quadratic and convex function of $\alpha_r$, therefore it has a unique minimum. While it is possible to compute the optimal value of $\alpha_r$ from Eq.5, here we just note that the loss is a sum of two quadratic functions, one has a minimum at $\alpha_r = 0$ and another has a minimum at $\alpha_r = 1/\left(1 + (p+1)/n_r\right)$, therefore the optimal learning rate is in between the two values and is always positive. This is intuitive, since a positive learning rate for adaptation implies that the parameters get closer to the optimum for the target task. An example of the loss as a function of the adaptation learning rate $\alpha_r$ is shown in Figure 2a, where we also show the results of experiments

in which we run MAML empirically. The good agreement between theory and experiment suggest that Eq.5 is accurate.

However, the training learning rate $\alpha_t$ shows the opposite: by taking the derivative of Eq.5 with respect to $\alpha_t$, it is possible to show that it has a unique absolute minimum for a negative value of $\alpha_t$. This can be proved by noting that this function has the same finite value for large positive or negative $\alpha_t$, its derivative is always positive at $\alpha_t = 0$, and it has one minimum $(-)$ and one maximum $(+)$ at values

$$\alpha_t^\pm = -\frac{n_t + 1}{2p} \pm \sqrt{\left(\frac{n_t + 1}{2p}\right)^2 + \frac{n_t}{p}} \tag{8}$$

Note that the argmax $\alpha_t^+$ is always positive, while the argmin $\alpha_t^-$ is always negative. This result is counter-intuitive, since a negative learning rate pushes parameters towards higher values of the loss. However, learning of the meta-parameter $\boldsymbol{\omega}$ is performed by the outer loop (minimize Eq.1), for which there is no learning rate since we are using the exact solution to the linear problem and thus we are effectively training to convergence. Therefore, it remains unclear whether the inner loop (Eq.3) should push parameters towards higher or lower values of the loss. An example of the loss as a function of the training learning rate $\alpha_r$ is shown in Figure 2b, where we also show the results of experiments in which we run MAML empirically. Here the theory slightly underestimate the experimental loss, but the overall shapes of the curves are in good agreement, suggesting that Eq.5 is accurate. Additional experiments are shown in the Appendix, Figure 6.

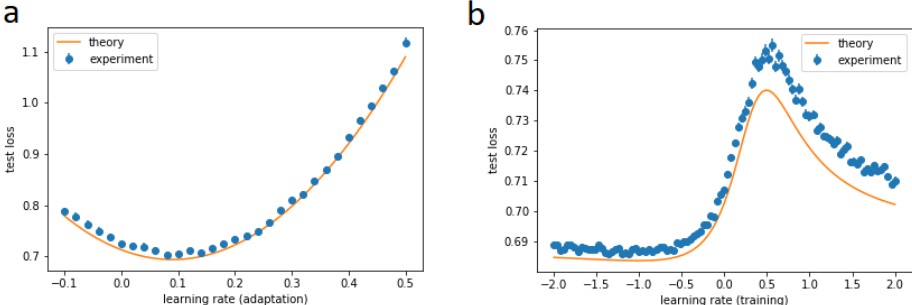

Figure 2: Average test loss of MAML as a function of the learning rate, on overparameterized mixed linear regression, as predicted by our theory and confirmed in experiments. a) Effect of learning rate $\alpha_r$ during adaptation. b) Effect of learning rate $\alpha_t$ during training. The optimal learning rate during adaptation is positive, while that during training is negative. Values of parameters: $n_t = 30, n_v = 2, n_r = 20, m = 3, p = 60, \sigma = 1., \nu = 0.5, \boldsymbol{\omega}_0 = \mathbf{0}, \mathbf{w}_0 = \mathbf{0}$. In panel a) we set $\alpha_t = 0.2$, in panel b) we set $\alpha_r = 0.2$. In the experiments, each run is evaluated on 100 test tasks of 50 data points each, and each point is an average over 100 runs (a) or 1000 runs (b).

## 5.2 Underparameterized case

In the underparameterized case, the number of parameters $p$ is smaller than the total number of validation data across tasks $n_v m$. In this case, since the data fully constrains the parameters, the optimal value of $\boldsymbol{\omega}$ found during meta-training is unique. We prove the following result.

**Theorem 2.** *Consider the algorithm of section 3 (MAML one-step), and the data generating model of section 4 (mixed linear regression). Let $p < n_v m$. Let $n_v(\xi)$ and $n_t(\xi)$ be any function of order $O(\xi)$. For $\xi, m \to \infty$, the test loss of Eq.2, averaged over the entire data distribution (see Eq.27 in the Appendix) is equal to*

$$\overline{\mathcal{L}}^{test} = \frac{\sigma^2}{2} \left(1 + \frac{\alpha_r^2 p}{n_r}\right) + \frac{h^r \nu^2}{2} +$$
$$+ \frac{h^r}{2h^{t2}} \frac{p}{n_v m} \left\{ \sigma^2 \left[ h^t + \frac{\alpha_t^2}{n_t} \left[(n_v + 1) g_1 + p g_2\right]\right] + \frac{\nu^2}{p} \left[(n_v + 1) g_3 + p g_4\right] \right\} + O\left((m\xi)^{-3/2}\right)$$

$$(9)$$

*where $h^r, h^t$ are defined as in previous section, Eqs.6, 7, and $g_i$ are order $O(1)$ polynomials in $\alpha_t$, see Eqs.98-101 in the Appendix.*

*Proof.* The proof of this Theorem can be found in the Appendix, sections 7.3, 7.3.2. □

Again, the loss always increases with the output noise $\sigma$ and task variability $\nu$. Furthermore, in this case the loss always decreases with the number of data points $n_v$, $n_r$, and tasks $m$. Note that, for a very large number of tasks $m$, the loss does not depend on meta-training hyperparameters $\alpha_t$, $n_v$, $n_t$. When the number of tasks is infinite, it doesn't matter whether we run the inner loop, and how much data we have for each task.

As in the overparameterized case, the loss is a quadratic and convex function of the adaptation learning rate $\alpha_r$, and there is a unique minimum. While the value of the argmin is different, in this case as well the loss is a sum of two quadratic functions, one with minimum at $\alpha_r = 0$ and another with a minimum at $\alpha_r = 1/(1 + (p + 1)/n_r)$, therefore the optimal learning rate is again in between the same two values and is always positive. Similar comments applies to this case: a positive learning rate for adaptation implies that the parameters get closer to the optimum for the target task. An example of the loss as a function of the adaptation learning rate $\alpha_r$ is shown in Figure 3a, where we also show the results of experiments in which we run MAML empirically. The good agreement between theory and experiment suggest that Eq.9 is accurate.

As a function of the training learning rate $\alpha_t$, the loss Eq.9 is the ratio of two fourth order polynomials, therefore it is not straightforward to determine its behaviour. However, it is possible to show that the following holds

$$\left. \frac{\partial \overline{\mathcal{L}}^{test}}{\partial \alpha_t} \right|_{\alpha_t=0} = \frac{\sigma^2 p}{n_v m} \geq 0 \tag{10}$$

suggesting that performance is always better for negative values of $\alpha_t$ around zero. Even if counter-intuitive, this finding aligns with that of previous section, and similar comments apply. An example of the loss as a function of the training learning rate $\alpha_r$ is shown in Figure 3b, where we also show the results of experiments in which we run MAML empirically. A good agreement is observed between theory and experiment, again suggesting that Eq.9 is accurate. Additional experiments are shown in the Appendix, Figure 6.

### 5.3 NON-GAUSSIAN THEORY IN OVERPARAMETERIZED MODELS

In previous sections we studied the performance of MAML applied to the problem of mixed linear regression. It remains unclear whether the results in the linear case are relevant for the more interesting case of nonlinear problems. Inspired by recent theoretical work, we consider the case of nonlinear regression with squared loss

$$\mathcal{L}(\boldsymbol{\omega}) = \underset{\mathbf{x}}{\mathbb{E}} \, \underset{y|\mathbf{x}}{\mathbb{E}} \, \frac{1}{2} \left[y - f(\mathbf{x}, \boldsymbol{\omega})\right]^2 \tag{11}$$

where $y$ is a target output and $f(\mathbf{x}, \boldsymbol{\omega})$ the output of a neural network with input $\mathbf{x}$ and parameters $\boldsymbol{\omega}$. The introduction of the Neural Tangent Kernel showed that, in the limit of infinitely wide neural networks, the output is a linear function of its parameters during the entire course of training (Jacot et al. (2018), Lee et al. (2019)). This is expressed by a first order Taylor expansion

$$f(\mathbf{x}, \boldsymbol{\omega}) \simeq f(\mathbf{x}, \boldsymbol{\omega}_0) + \mathbf{k}(\mathbf{x}, \boldsymbol{\omega}_0)^T (\boldsymbol{\omega} - \boldsymbol{\omega}_0) \tag{12}$$

$$\mathbf{k}(\mathbf{x}, \boldsymbol{\omega}_0) = \nabla_{\boldsymbol{\omega}} f(\mathbf{x}, \boldsymbol{\omega})|_{\mathbf{x}, \boldsymbol{\omega}_0} \tag{13}$$

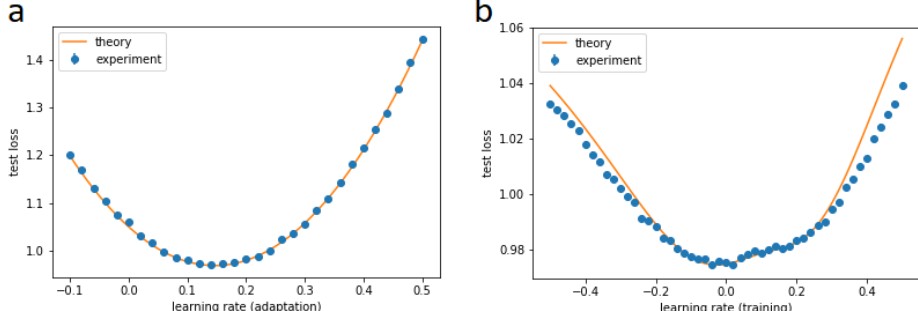

Figure 3: Average test loss as a function of the learning rate, on underparameterized mixed linear regression, as predicted by our theory and confirmed in experiments. a) Effect of learning rate $\alpha_r$ during testing. b) Effect of learning rate $\alpha_t$ during training. The optimal learning rate during testing is always positive, while that during training is negative. Values of parameters: $n_t = 5, n_v = 25, n_r = 10, m = 40, p = 30, \sigma = 0.2, \nu = 0.2$. In panel a) we set $\alpha_t = 0.2$, in panel b) we set $\alpha_r = 0.2$. In the experiments, the model is evaluated on 100 tasks of 50 data points each, and each point is an average over 100 (a) or 1000 (b) runs.

The parameters $\omega$ remain close to the initial condition $\omega_0$ during the entire course of training, a phenomenon referred to as *lazy training* (Chizat et al. (2020)), and therefore the output can be linearized around $\omega_0$. Intuitively, in a model that is heavily overparameterized, the data does not constrain the parameters, and a parameter that minimizes the loss in Eq.11 can be found in the vicinity of any initial condition $\omega_0$. Note that, while the output of the neural network is linear in the parameters, it remains a nonlinear function of its input, through the vector of nonlinear functions $\mathbf{k}$ in Eq.13.

By substituting Eq.12 into Eq.11, the nonlinear regression becomes effectively linear, in the sense that the loss is a quadratic function of the parameters $\omega$, and all nonlinearities are contained in the functions $\mathbf{k}$ in Eq.13, that are fixed by the initial condition $\omega_0$. This suggests that we can carry over the theory developed in the previous section to this problem. However, in this case the input to the linear regression problem is effectively $\mathbf{k}(\mathbf{x})$, and some of the assumptions made in the previous section are not acceptable. In particular, even if we assume that $\mathbf{x}$ is Gaussian, $\mathbf{k}(\mathbf{x})$ is a nonlinear function of $\mathbf{x}$ and cannot be assumed Gaussian. We prove the following result, where we generalize the result of section 5.1 to non-Gaussian inputs and weights.

**Theorem 3.** *Consider the algorithm of section 3 (MAML one-step), with $\omega_0 = 0$, and the data generating model of section 4, where the input $\mathbf{x}$ and the weights $\mathbf{w}$ are not necessarily Gaussian, and have zero mean and covariances, respectively, $\Sigma = \mathbb{E}\mathbf{x}\mathbf{x}^T$ and $\Sigma_w = \mathbb{E}\mathbf{w}\mathbf{w}^T$. Let $F$ be the matrix of fourth order moments $F = \mathbb{E}\left(\mathbf{x}^T \Sigma \mathbf{x}\right) \mathbf{x}\mathbf{x}^T$. Let $p > n_v m$. Let $p(\xi)$ and $n_t(\xi)$ be any function of order $O(\xi)$ as $\xi \to \infty$. Let $Tr\left(\Sigma_w^2\right)$ be of order $O\left(\xi^{-1}\right)$, and let the variances of matrix products of the rescaled inputs $\mathbf{x}/\sqrt{p}$, up to sixth order, be of order $O\left(\xi^{-1}\right)$ (see Eqs.134-136 in the Appendix). Then the test loss of Eq.2, averaged over the entire data distribution (see Eq.27 in the Appendix) is equal to*

$$\overline{\mathcal{L}}^{test} = \frac{1}{2}Tr\left(\Sigma_w H^r\right) + \frac{\sigma^2}{2}\left[1 + \frac{\alpha_r^2}{n_r}Tr\left(\Sigma^2\right)\right] +$$

$$+ \frac{1}{2}n_v m \frac{Tr\left(H^r H^t\right)\left\{Tr\left(\Sigma_w H^t\right) + \sigma^2\left[1 + \frac{\alpha_t^2}{n_t}Tr\left(\Sigma^2\right)\right]\right\}}{Tr\left(H^t\right)^2} + O\left(\xi^{-3/2}\right) \quad (14)$$

*where we define the following matrices*

$$H^t = \left[\Sigma\left(I - \alpha_t\Sigma\right)^2 + \frac{\alpha_t^2}{n_t}\left(F - \Sigma^3\right)\right] \quad (15)$$

$$H^r = \left[\Sigma\left(I - \alpha_r\Sigma\right)^2 + \frac{\alpha_r^2}{n_r}\left(F - \Sigma^3\right)\right] \quad (16)$$

*Proof.* The proof of this Theorem can be found in Appendix, section 7.4. □

Note that this result reduces to Eqs.5, 6, 7 when $\Sigma = I$, $\Sigma_w = I\nu^2/p$, $F = I(p+2)$, $\omega_0 = \mathbf{0}$, $\mathbf{w} = \mathbf{0}$. This expression for the loss is more difficult to analyze than those given in the previous sections, because it involves traces of nonlinear functions of matrices, all elements of which are free hyperparameters. Nevertheless, it is possible to show that, as a function of the adaptation learning rate $\alpha_r$, the loss in Eq.14 is still a quadratic function. As a function of the adaptation learning rate $\alpha_r$, the loss in Eq.14 is the ratio of two fourth order polynomials, but it is difficult to draw any conclusions since their coefficients do not appear to have simple relationships.

Even if the influence of the hyperparameters is not easy to predict, the expression in Eq.14 can still be used to quickly probe the behavior of the loss empirically, by using example values for the $\Sigma$, $\Sigma_w$, $F$, since computing the expression is very fast. Here we choose values of $\Sigma$, $\Sigma_w$ by a single random draw from a Wishart distribution

$$\Sigma \sim \mathcal{W}(I, p) \qquad \Sigma_w \sim \frac{\nu^2}{p}\mathcal{W}(I, p) \tag{17}$$

Note that the number of degrees of freedom of the distribution is equal to the size of the matrices, $p$, therefore this covariances display significant correlations. Furthermore, we choose $F = 2\Sigma^3 + \Sigma\text{Tr}(\Sigma^2)$, which is the value taken when $\mathbf{x}$ follows a Gaussian distribution. Therefore, we effectively test the loss in Eq.14 for a Gaussian distribution, as in previous section, but we stress that the expression is valid for any distribution of $\mathbf{x}$ within the assumptions of Theorem 3. We also run experiments of MAML, applied again to mixed linear regression, but now using the covariance matrices drawn in Eq.17. Figure 4 shows the loss in Eq.14 as a function of the learning rates, during adaptation (panel a) and training (panel b). Qualitatively, we observe a similar behaviour as in section 5.1: the adaptation learning rate has a unique minimum for a positive value of $\alpha_r$, while the training learning rate shows better performance for negative values of $\alpha_t$. Again, there is a good agreement between theory and experiment, suggesting that Eq.14 is a good approximation.

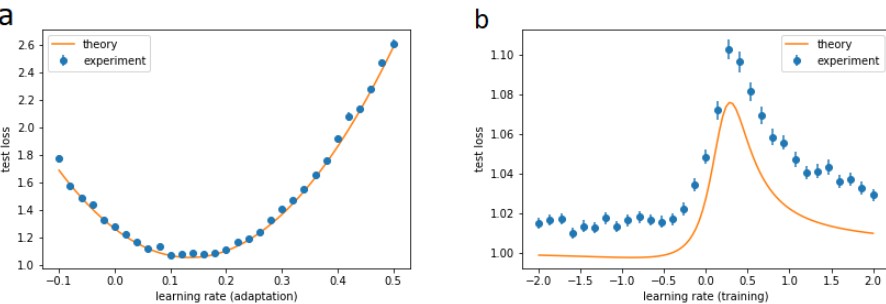

Figure 4: Average test loss of MAML as a function of the learning rate, on overparameterized mixed linear regression with Wishart covariances, as predicted by our theory and confirmed in experiments. a) Effect of learning rate $\alpha_r$ during adaptation. b) Effect of learning rate $\alpha_t$ during training. The optimal learning rate during adaptation is positive, while that during training appears to be negative. Values of parameters: $n_t = 30, n_v = 2, n_r = 20, m = 3, p = 60, \sigma = 1., \nu = 0.5, \omega_0 = \mathbf{0}$, $\mathbf{w}_0 = \mathbf{0}$. In panel a) we set $\alpha_t = 0.2$, in panel b) we set $\alpha_r = 0.2$. In the experiments, each run is evaluated on 100 tasks of 50 data points each, and each point is an average over 100 runs (a) or 500 runs (b).

## 5.4 NONLINEAR REGRESSION

To investigate whether negative learning rates improve performance on non-linear regression in practice, we studied the simple case of MAML with a neural network applied to a quadratic function. Specifically, the target output is generated according to $y = (\mathbf{w}^T\mathbf{x} + b)^2 + z$, where $b$ is a bias term. The data $\mathbf{x}$, $z$ and generating parameters $\mathbf{w}$ are sampled as described in section 4 (in addition, the bias $b$ was drawn from a Gaussian distribution of zero mean and unit variance.). We use a 2-layer

feed-forward neural network with ReLU activation functions. Weights are initialized following a Gaussian distribution of zero mean and variance equal to the inverse number of inputs. We report results with a network width of $400$ in both layers; results were similar with larger network widths. We use the square loss function and we train the neural network in the outer loop with stochastic gradient descent with a learning rate of $0.001$ for $5000$ epochs (until convergence). We used most parameters identical to section 5.1: $n_t = 30; n_v = 2; n_r = 20; m = 3; p = 60; \sigma = 1, \nu = 0.5, \mathbf{w}_0 = 0$. The learning rate for adaptation was set to $\alpha_r = 0.01$. Note that in section 5.1 the model was initialized at the ground truth of the generative model ($\boldsymbol{\omega}_0 = \mathbf{w}_0$), while here the neural network parameters are initialized at random. Figure 5 shows the test loss as a function of the learning rate $\alpha_t$. The best performance is obtained for a negative learning rate of $\alpha_t = -0.0075$.

## 6  DISCUSSION

We calculated algebraic expressions for the average test loss of MAML applied to a simple family of linear models, as a function of the hyperparameters. Surprisingly, we showed that the optimal value of the learning rate of the inner loop during training is negative. This finding seems to carry over to more interesting nonlinear models in the overparameterized case. However, additional work is necessary to establish the conditions under which the optimal learning rate may be positive, for example by probing more extensively Eq.14.

A negative optimal learning rate is surprising and counter-intuitive, since negative learning rates push parameters towards higher values of the loss. However, the meta-training loss is minimized by the outer loop, therefore it is not immediately obvious whether the learning rate of the inner loop should be positive, and we show that in some circumstances it should not. However, perhaps obviously, we also show that the learning rate during adaptation at test time should always be positive, otherwise the target task cannot be learned.

In this work, we considered the case of nonlinear models in the overparameterized case. However, typical applications of MAML (and meta-learning in general) implement relatively small models due to the heavy computational load of running bi-level optimization, including both outer and inner loop. Our theory applies to regression problems, and assumes a limited number of tasks where data is independently drawn in each task, while some applications use a large number of tasks with correlated draws (for example, images may be shared across tasks in few-shot image classification, see Bertinetto et al. (2019)). Our theory is valid at the exact optimum of the outer loop, which is equivalent to training the outer loop to convergence, therefore overfitting may occur in the outer loop of our model. Another limitation of our theory is represented by the assumptions on the input and task covariance, which have no correlations in Theorems 1, 2, and are subject to some technical assumptions in Theorem 3.

To the best of our knowledge, nobody has considered training meta-learning models with negative learning rates in the inner loop. Given that some studies advocate removing the inner loop altogether, which is similar to setting the learning rate to zero, it would be interesting to try a negative one. On the other hand, it is possible that a negative learning rate does not work in classification problems, in nonlinear models, or using input or tasks with a complex structure, settings that are outside the theory presented in this work.

We would like to thank Paolo Grazieschi for helping with formalizing the theorems, and Ritwik Niyogi for helping with nonlinear regression experiments.

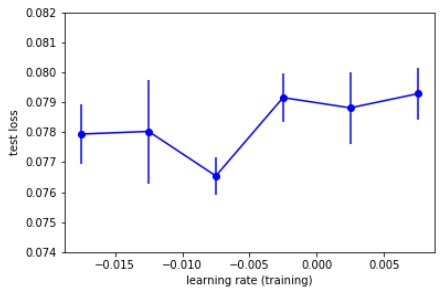

Figure 5: Average test loss of MAML as a function of the learning rate, on nonlinear (quadratic) regression using a 2-layer feed-forward neural network. Optimal learning rate is negative, consistent with results on the linear case. Each run is evaluated on 1000 test tasks, and each point is an average over 10 runs. Error bars show standard errors. Note the qualitative similarity with Figures 2b and 4b.

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

# 7 APPENDIX

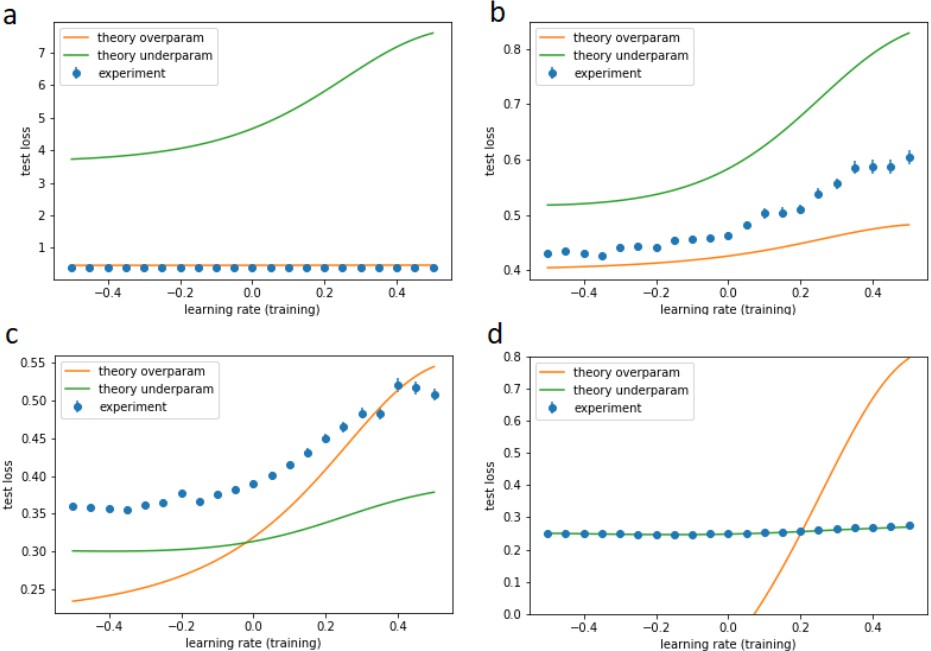

Figure 6: Average test loss of MAML as a function of the learning rate $\alpha_t$ (training) on mixed linear regression, showing the transition from strongly overparameterized (a), to weakly overparameterized (b), weakly underparameterized (c) and strongly underparameterized (d). As expected, predictions of theory are accurate only in panels (a) and (d). The amount of validation data increases from panels (a) to (d), with the following values: $m = 1, n_v = 2$ (a), $m = 5, n_v = 5$ (b), $m = 10, n_v = 10$ (c), $m = 10, n_v = 40$. Other parameters are equal to: $n_t = 40, n_r = 40, p = 50, \sigma = 0.5., \nu = 0.5,$ $\alpha_r = 0.2, \boldsymbol{\omega}_0 = \mathbf{0}, \mathbf{w}_0 = (0.1, 0.1, \ldots, 0.1)$ (note that overfitting occurs since $\boldsymbol{\omega}_0 \neq \mathbf{w}_0$). In the experiments, each run is evaluated on 100 test tasks of 50 data points each, and each point is an average over 100 runs.

## 7.1 DEFINITION OF THE LOSS FUNCTION

We consider the problem of mixed linear regression $\mathbf{y} = X\mathbf{w} + \mathbf{z}$ with squared loss, where $X$ is a $n \times p$ matrix of input data, each row is one of $n$ data vectors of dimension $p$, $\mathbf{z}$ is a $n \times 1$ noise vector, $\mathbf{w}$ is a $p \times 1$ vector of generating parameters and $\mathbf{y}$ is a $n \times 1$ output vector. Data is collected for $m$ tasks, each with a different value of the parameters $\mathbf{w}$ and a different realization of the input $X$ and noise $\mathbf{z}$. We denote by $\mathbf{w}^{(i)}$ the parameters for task $i$, for $i = 1, \ldots, m$. For a given task $i$, we denote by $X^{t(i)}, X^{v(i)}$ the input data for, respectively, the training and validation sets, by $\mathbf{z}^{t(i)}, \mathbf{z}^{v(i)}$ the corresponding noise vectors and by $\mathbf{y}^{t(i)}, \mathbf{y}^{v(i)}$ the output vectors. We denote by $n_t, n_v$ the data sample size for training and validations sets, respectively.

For a given task $i$, the training output is equal to

$$\mathbf{y}^{t(i)} = X^{t(i)}\mathbf{w}^{(i)} + \mathbf{z}^{t(i)} \tag{18}$$

Similarly, the validation output is equal to

$$\mathbf{y}^{v(i)} = X^{v(i)}\mathbf{w}^{(i)} + \mathbf{z}^{v(i)}. \tag{19}$$

We consider MAML as a model for meta-learning (Finn et al 2017). The meta-training loss is equal to

$$\mathcal{L}^{meta} = \frac{1}{2n_v m} \sum_{i=1}^{m} \left| \mathbf{y}^{v(i)} - X^{v(i)}\boldsymbol{\theta}^{(i)}(\boldsymbol{\omega}) \right|^2 \tag{20}$$

where vertical brackets denote euclidean norm, and the estimated parameters $\boldsymbol{\theta}^{(i)}(\boldsymbol{\omega})$ are equal to the one-step gradient update on the single-task training loss $\mathcal{L}^{(i)} = |\mathbf{y}^{t(i)} - X^{t(i)}\boldsymbol{\theta}^{(i)}|^2/2n_t$, with initial condition given by the meta-parameter $\boldsymbol{\omega}$. The single gradient update is equal to

$$\boldsymbol{\theta}^{(i)}(\boldsymbol{\omega}) = \left(I_p - \frac{\alpha_t}{n_t}X^{t(i)T}X^{t(i)}\right)\boldsymbol{\omega} + \frac{\alpha_t}{n_t}X^{t(i)T}\mathbf{y}^{t(i)} \tag{21}$$

where $I_p$ is the $p \times p$ identity matrix and $\alpha_t$ is the learning rate. We seek to minimize the meta-training loss with respect to the meta-parameter $\boldsymbol{\omega}$, namely

$$\boldsymbol{\omega}^\star = \arg\min_{\boldsymbol{\omega}} \mathcal{L}^{meta} \tag{22}$$

We evaluate the solution $\boldsymbol{\omega}^\star$ by calculating the meta-test loss

$$\mathcal{L}^{test} = \frac{1}{2n_s}|\mathbf{y}^s - X^s\boldsymbol{\theta}^\star|^2 \tag{23}$$

Note that the test loss is calculated over test data $X^s, \mathbf{z}^s$, and test parameters $\mathbf{w}'$, namely

$$\mathbf{y}^s = X^s\mathbf{w}' + \mathbf{z}^s \tag{24}$$

Furthermore, the estimated parameters $\boldsymbol{\theta}^\star$ are calculated on a separate set of target data $X^r, \mathbf{z}^r$, namely

$$\boldsymbol{\theta}^\star = \left(I_p - \frac{\alpha_r}{n_r}X^{rT}X^r\right)\boldsymbol{\omega}^\star + \frac{\alpha_r}{n_r}X^{rT}\mathbf{y}^r \tag{25}$$

$$\mathbf{y}^r = X^r\mathbf{w}' + \mathbf{z}^r \tag{26}$$

Note that the learning rate and sample size can be different at testing, denoted by $\alpha_r, n_r, n_s$. We are interested in calculating the average test loss, that is the test loss of Eq.23 averaged over the entire data distribution, equal to

$$\overline{\mathcal{L}}^{test} = \mathbb{E}_{\mathbf{w}}\mathbb{E}_{\mathbf{z}^t}\mathbb{E}_{X^t}\mathbb{E}_{\mathbf{z}^v}\mathbb{E}_{X^v}\mathbb{E}_{\mathbf{w}'}\mathbb{E}_{\mathbf{z}^s}\mathbb{E}_{X^s}\mathbb{E}_{\mathbf{z}^r}\mathbb{E}_{X^r} \frac{1}{2n_s}|\mathbf{y}^s - X^s\boldsymbol{\theta}^\star|^2 \tag{27}$$

## 7.2 Definition of probability distributions

We assume that all random variables are Gaussian. In particular, we assume that the rows of the matrix $X$ are independent, and each row, denoted by $\mathbf{x}$, is distributed according to a multivariate Gaussian with zero mean and unit covariance

$$\mathbf{x} \sim \mathcal{N}(0, I_p) \tag{28}$$

where $I_p$ is the $p \times p$ identity matrix. Similarly, the noise is distributed following a multivariate Gaussian with zero mean and variance equal to $\sigma^2$, namely

$$\mathbf{z} \sim \mathcal{N}(0, \sigma^2 I_n) \tag{29}$$

Finally, the generating parameters are also distributed according to a multivariate Gaussian of variance $\nu^2/p$, namely

$$\mathbf{w} \sim \mathcal{N}\left(\mathbf{w}_0, \frac{\nu^2}{p}I_p\right) \tag{30}$$

The generating parameter $\mathbf{w}$ is drawn once and kept fixed within a task, and drawn independently for different tasks. The values of $\mathbf{x}$ and $\mathbf{z}$ are drawn independently in all tasks and datasets (training, validation, target, test). In order to perform the calculations in the next section, we need the following results.

**Lemma 1.** *Let $X$ be a Gaussian $n \times p$ random matrix with independent rows, and each row has covariance equal to $I_p$, the $p \times p$ identity matrix. Then:*

$$\mathbb{E}\left[X^T X\right] = n I_p \tag{31}$$

$$\mathbb{E}\left[\left(X^T X\right)^2\right] = n\left(n + p + 1\right) I_p = n^2 \mu_2 I_p \tag{32}$$

$$\mathbb{E}\left[\left(X^T X\right)^3\right] = n\left(n^2 + p^2 + 3np + 3n + 3p + 4\right) I_p = n^3 \mu_3 I_p \tag{33}$$

$$\mathbb{E}\left[\left(X^T X\right)^4\right] = n\left(n^3 + p^3 + 6n^2 p + 6np^2 + \right. \tag{34}$$

$$\left. +6n^2 + 6p^2 + 17np + 21n + 21p + 20\right) I_p = n^4 \mu_4 I_p \tag{35}$$

$$\mathbb{E}\left[X^T X \, Tr\left(X^T X\right)\right] = \left(n^2 p + 2n\right) I_p = pn^2 \mu_{1,1} I_p \tag{36}$$

$$\mathbb{E}\left[\left(X^T X\right)^2 Tr\left(X^T X\right)\right] = n\left(n^2 p + np^2 + np + 4n + 4p + 4\right) I_p = pn^3 \mu_{2,1} I_p \tag{37}$$

$$\mathbb{E}\left[X^T X \, Tr\left(\left(X^T X\right)^2\right)\right] = n\left(n^2 p + np^2 + np + 4n + 4p + 4\right) I_p = pn^3 \mu_{1,2} I_p \tag{38}$$

$$\mathbb{E}\left[\left(X^T X\right)^2 Tr\left(\left(X^T X\right)^2\right)\right] = n\left(n^3 p + np^3 + 2n^2 p^2 + 2n^2 p + 2np^2 + \right. \tag{39}$$

$$\left. +8n^2 + 8p^2 + 21np + 20n + 20p + 20\right) I_p = pn^4 \mu_{2,2} I_p \tag{40}$$

*where the last equality in each of these expressions defines the variables $\mu$. Furthermore, for any $n \times n$ symmetric matrix $C$ and any $p \times p$ symmetric matrix $D$, independent of $X$:*

$$\mathbb{E}\left[X^T C X\right] = Tr\left(C\right) I_p \tag{41}$$

$$\mathbb{E}\left[X^T X D X^T X\right] = n\left(n + 1\right) D + n Tr\left(D\right) I_p \tag{42}$$

*Proof.* The Lemma follows by direct computations of the above expectations, using Isserlis' theorem. Particularly, for higher order exponents, combinatorics plays a crucial role in counting products of different Gaussian variables in an effective way.

$\square$

**Lemma 2.** *Let $X^{v(i)}$, $X^{t(i)}$ be Gaussian random matrices, of size respectively $n_v \times p$ and $n_t \times p$, with independent rows, and each row has covariance equal to $I_p$, the $p \times p$ identity matrix. Let $p(\xi)$ and $n_t(\xi)$ be any function of order $O(\xi)$ as $\xi \to \infty$. Then:*

$$X^{v(i)} X^{v(i)T} = p \, I_{n_v} + O\left(\xi^{1/2}\right) \tag{43}$$

$$X^{v(i)} X^{t(i)T} X^{t(i)} X^{v(i)T} = pn_t \, I_{n_v} + O\left(\xi^{3/2}\right) \tag{44}$$

$$X^{v(i)} X^{t(i)T} X^{t(i)} X^{t(i)T} X^{t(i)} X^{v(i)T} = pn_t(n_t + p + 1) I_{n_v} + O\left(\xi^{5/2}\right) \tag{45}$$

*Note that the order $O\left(\xi\right)$ applies to all elements of the matrix in each expression. For $i \neq j$*

$$X^{v(i)} X^{v(j)T} = O\left(\xi^{1/2}\right) \tag{46}$$

$$X^{v(i)} X^{t(i)T} X^{t(i)} X^{v(j)T} = O\left(\xi^{3/2}\right) \tag{47}$$

$$X^{v(i)} X^{t(i)T} X^{t(i)} X^{t(j)T} X^{t(j)} X^{v(j)T} = O\left(\xi^{5/2}\right) \tag{48}$$

*Furthermore, for any positive real number $\delta$ and for any $p \times p$ symmetric matrix $D$ independent of $X$, where $Tr(D)$ and $Tr(D^2)$ are both of order $O(\xi^\delta)$*

$$X^{v(i)} D X^{v(i)^T} = Tr(D) I_{n_v} + O\left(\xi^{\delta/2}\right) \tag{49}$$

$$X^{v(i)} X^{t(i)^T} X^{t(i)} D X^{v(i)^T} = Tr(D) n_t I_{n_v} + O\left(\xi^{1+\delta/2}\right) \tag{50}$$

$$X^{v(i)} X^{t(i)^T} X^{t(i)} D X^{t(i)^T} X^{t(i)} X^{v(i)^T} = Tr(D) n_t(n_t + p + 1) I_{n_v} + O\left(\xi^{2+\delta/2}\right) \tag{51}$$

$$X^{v(i)} D X^{v(j)^T} = O\left(\xi^{\delta/2}\right) \tag{52}$$

$$X^{v(i)} X^{t(i)^T} X^{t(i)} D X^{v(j)^T} = O\left(\xi^{1+\delta/2}\right) \tag{53}$$

$$X^{v(i)} X^{t(i)^T} X^{t(i)} D X^{t(j)^T} X^{t(j)} X^{v(j)^T} = O\left(\xi^{2+\delta/2}\right) \tag{54}$$

*Proof.* The Lemma follows by direct computations of the expectations and variances of each term.

$\square$

**Lemma 3.** *Let $X^v$, $X^t$ be Gaussian random matrices, of size respectively $n_v \times p$ and $n_t \times p$, with independent rows, and each row has covariance equal to $I_p$, the $p \times p$ identity matrix. Let $n_v(\xi)$ and $n_t(\xi)$ be any function of order $O(\xi)$ for $\xi \to \infty$. Then:*

$$X^{v^T} X^v = n_v I_p + O\left(\xi^{1/2}\right) \tag{55}$$

$$X^{t^T} X^t X^{v^T} X^v = n_t n_v I_p + O\left(\xi^{3/2}\right) \tag{56}$$

$$X^{t^T} X^t X^{v^T} X^v X^{t^T} X^t = n_v n_t(n_t + p + 1) I_p + O\left(\xi^{5/2}\right) \tag{57}$$

*Note that the order $O(\xi)$ applies to all elements of the matrix in each expression.*

*Proof.* The Lemma follows by direct computations of the expectations and variances of each term.

$\square$

## 7.3 PROOF OF THEOREMS 1 AND 2

We calculate the average test loss as a function of the hyperparameters $n_t$, $n_v$, $n_r$, $p$, $m$, $\alpha_t$, $\alpha_r$, $\sigma$, $\nu$, $\mathbf{w}_0$. Using the expression in Eq.24 for the test output, we rewrite the test loss in Eq.27 as

$$\overline{\mathcal{L}}^{test} = \mathbb{E} \frac{1}{2n_s} |X^s(\mathbf{w}' - \boldsymbol{\theta}^\star) + \mathbf{z}^s|^2 \tag{58}$$

We start by averaging this expression with respect to $X^s, \mathbf{z}^s$, noting that $\boldsymbol{\theta}^\star$ does not depend on test data. We further average with respect to $\mathbf{w}'$, but note that $\boldsymbol{\theta}^\star$ depends on test parameters, so we average only terms that do not depend on $\boldsymbol{\theta}^\star$. Using Eq.31, the result is

$$\overline{\mathcal{L}}^{test} = \frac{\sigma^2}{2} + \frac{\nu^2}{2} + \frac{|\mathbf{w}_0|^2}{2} + \mathbb{E}\left[\frac{|\boldsymbol{\theta}^\star|^2}{2} - (\mathbf{w}_0 + \delta\mathbf{w}')^T \boldsymbol{\theta}^\star\right] \tag{59}$$

where we define $\delta\mathbf{w}' = \mathbf{w}' - \mathbf{w}_0$. The second term in the expectation is linear in $\boldsymbol{\theta}^\star$ and can be averaged over $X^r, \mathbf{z}^r$, using Eq.25 and noting that $\boldsymbol{\omega}^\star$ does not depend on target data. The result is

$$\mathbb{E}_{X^r} \mathbb{E}_{\mathbf{z}^r} \boldsymbol{\theta}^\star = (1 - \alpha_r)\boldsymbol{\omega}^\star + \alpha_r (\mathbf{w}_0 + \delta\mathbf{w}') \tag{60}$$

Using Eq.60 we average over $\mathbf{w}'$ the second term in the expectation of Eq.59 and find

$$\overline{\mathcal{L}}^{test} = \frac{\sigma^2}{2} + \left(\frac{1}{2} - \alpha_r\right)\left(\nu^2 + |\mathbf{w}_0|^2\right) - (1 - \alpha_r)\mathbf{w}_0^T \mathbb{E}\, \boldsymbol{\omega}^\star + \mathbb{E}\frac{|\boldsymbol{\theta}^\star|^2}{2} \tag{61}$$

We average the last term of this expression over $\mathbf{z}^r, \mathbf{w}'$, using Eq.25 and noting that $\boldsymbol{\omega}^\star$ does not depend on target data and test parameters. The result is

$$\mathop{\mathbb{E}}_{\mathbf{w}'} \mathop{\mathbb{E}}_{\mathbf{z}^r} |\boldsymbol{\theta}^\star|^2 = |\boldsymbol{\omega}^\star|^2 + \frac{\alpha_r^2}{n_r^2} (\boldsymbol{\omega}^\star - \mathbf{w}_0)^T \left(X^{r T} X^r\right)^2 (\boldsymbol{\omega}^\star - \mathbf{w}_0) - \tag{62}$$

$$- \frac{2\alpha_r}{n_r} X^{r T} X^r \boldsymbol{\omega}^{\star T} (\boldsymbol{\omega}^\star - \mathbf{w}_0) + \frac{\alpha_r^2 \sigma^2}{n_r^2} \mathrm{Tr} \left[X^r X^{r T}\right] + \frac{\alpha_r^2 \nu^2}{n_r^2 p} \mathrm{Tr} \left[\left(X^r X^{r T}\right)^2\right] \tag{63}$$

We now average over $X^r$, again noting that $\boldsymbol{\omega}^\star$ does not depend on target data. Using Eqs.31, 32, we find

$$\mathop{\mathbb{E}}_{X^r} \mathop{\mathbb{E}}_{\mathbf{w}'} \mathop{\mathbb{E}}_{\mathbf{z}^r} |\boldsymbol{\theta}^\star|^2 = |\boldsymbol{\omega}^\star|^2 + \alpha_r^2 \left(1 + \frac{p+1}{n_r}\right) \left(\nu^2 + |\boldsymbol{\omega}^\star - \mathbf{w}_0|^2\right) - 2\alpha_r \boldsymbol{\omega}^{\star T} (\boldsymbol{\omega}^\star - \mathbf{w}_0) + \frac{\alpha_r^2 \sigma^2 p}{n_r} \tag{64}$$

We can now rewrite the average test loss 61 as

$$\overline{\mathcal{L}}^{test} = \frac{\sigma^2}{2} \left(1 + \frac{\alpha_r^2 p}{n_r}\right) + \frac{1}{2} \left[(1 - \alpha_r)^2 + \alpha_r^2 \frac{p+1}{n_r}\right] \left(\nu^2 + \mathbb{E} |\boldsymbol{\omega}^\star - \mathbf{w}_0|^2\right) \tag{65}$$

In order to average the last term, we need an expression for $\boldsymbol{\omega}^\star$. We note that the loss in Eq.20 is quadratic in $\boldsymbol{\omega}$, therefore the solution of Eq.22 can be found using standard linear algebra. In particular, the loss in Eq.20 can be rewritten as

$$\mathcal{L}^{meta} = \frac{1}{2n_v m} |\boldsymbol{\gamma} - B\boldsymbol{\omega}|^2 \tag{66}$$

where $\boldsymbol{\gamma}$ is a vector of shape $n_v m \times 1$, and $B$ is a matrix of shape $n_v m \times p$. The vector $\boldsymbol{\gamma}$ is a stack of $m$ vectors

$$\boldsymbol{\gamma} = \begin{pmatrix} X^{v(1)} \left(I_p - \frac{\alpha_t}{n_t} X^{t(1)T} X^{t(1)}\right) \mathbf{w}^{(1)} - \frac{\alpha_t}{n_t} X^{v(1)} X^{t(1)T} \mathbf{z}^{t(1)} + \mathbf{z}^{v(1)} \\ \vdots \\ X^{v(m)} \left(I_p - \frac{\alpha_t}{n_t} X^{t(m)T} X^{t(m)}\right) \mathbf{w}^{(m)} - \frac{\alpha_t}{n_t} X^{v(m)} X^{t(m)T} \mathbf{z}^{t(m)} + \mathbf{z}^{v(m)} \end{pmatrix} \tag{67}$$

Similarly, the matrix $B$ is a stack of $m$ matrices

$$B = \begin{pmatrix} X^{v(1)} \left(I_p - \frac{\alpha_t}{n_t} X^{t(1)T} X^{t(1)}\right) \\ \vdots \\ X^{v(m)} \left(I_p - \frac{\alpha_t}{n_t} X^{t(m)T} X^{t(m)}\right) \end{pmatrix} \tag{68}$$

We denote by $I_p$ the $p \times p$ identity matrix. The expression for $\boldsymbol{\omega}$ that minimizes Eq.66 depends on whether the problem is overparameterized ($p > n_v m$) or underparameterized ($p < n_v m$), therefore we distinguish these two cases in the following sections.

### 7.3.1 OVERPARAMETERIZED CASE (THEOREM 1)

In the overparameterized case ($p > n_v m$), under the assumption that the inverse of $BB^T$ exists, the value of $\boldsymbol{\omega}$ that minimizes Eq.66 is equal to

$$\boldsymbol{\omega}^\star = B^T \left(BB^T\right)^{-1} \boldsymbol{\gamma} + \left[I_p - B^T \left(BB^T\right)^{-1} B\right] \boldsymbol{\omega}_0 \tag{69}$$

The vector $\boldsymbol{\omega}_0$ is interpreted as the initial condition of the parameter optimization of the outer loop, when optimized by gradient descent. Note that the matrix $B$ does not depend on $\mathbf{w}, \mathbf{z}^t, \mathbf{z}^v$, and $\mathbb{E}_{\mathbf{w}} \mathbb{E}_{\mathbf{z}^t} \mathbb{E}_{\mathbf{z}^v} \boldsymbol{\gamma} = B\mathbf{w}_0$. We denote by $\delta\boldsymbol{\gamma}$ the deviation from the average, and we have

$$\boldsymbol{\omega}^\star - \mathbf{w}_0 = B^T \left(BB^T\right)^{-1} \delta\boldsymbol{\gamma} + \left[I_p - B^T \left(BB^T\right)^{-1} B\right] (\boldsymbol{\omega}_0 - \mathbf{w}_0) \tag{70}$$

We square this expression and average over $\mathbf{w}, \mathbf{z}^t, \mathbf{z}^v$. We use the cyclic property of the trace and the fact that $B^T \left(BB^T\right)^{-1} B$ is a projection. The result is

$$|\boldsymbol{\omega}^\star - \mathbf{w}_0|^2 = \mathrm{Tr} \left[\Gamma \left(BB^T\right)^{-1}\right] + (\boldsymbol{\omega}_0 - \mathbf{w}_0)^T \left[I_p - B^T \left(BB^T\right)^{-1} B\right] (\boldsymbol{\omega}_0 - \mathbf{w}_0) \tag{71}$$

The matrix $\Gamma$ is defined as

$$\Gamma = \underset{\mathbf{w}}{\mathbb{E}}\, \underset{\mathbf{z}^t}{\mathbb{E}}\, \underset{\mathbf{z}^v}{\mathbb{E}}\, \delta\boldsymbol{\gamma}\, \delta\boldsymbol{\gamma}^T = \begin{pmatrix} \Gamma^{(1)} & 0 & 0 \\ 0 & \ddots & 0 \\ 0 & 0 & \Gamma^{(m)} \end{pmatrix} \tag{72}$$

Where matrix blocks are given by the following expression

$$\Gamma^{(i)} = \frac{\nu^2}{p} X^{v(i)} \left( I_p - \frac{\alpha_t}{n_t} X^{t(i)\,T} X^{t(i)} \right)^2 X^{v(i)\,T} + \sigma^2 \left( I_{n_v} + \frac{\alpha_t^2}{n_t^2} X^{v(i)} X^{t(i)\,T} X^{t(i)} X^{v(i)\,T} \right) \tag{73}$$

It is convenient to rewrite the scalar product of Eq.71 in terms of the trace of outer products

$$|\boldsymbol{\omega}^\star - \mathbf{w}_0|^2 = \mathrm{Tr}\left[ (BB^T)^{-1} \left( \Gamma - B(\boldsymbol{\omega}_0 - \mathbf{w}_0)(\boldsymbol{\omega}_0 - \mathbf{w}_0)^T B^T \right) \right] + |\boldsymbol{\omega}_0 - \mathbf{w}_0|^2 \tag{74}$$

In order to calculate $\mathbb{E}\,|\boldsymbol{\omega}^\star - \mathbf{w}_0|^2$ in Eq.65 we need to average this expression over training and validation data. These averages are hard to compute since they involve nonlinear functions of the data. However, we can approximate these terms by assuming that $p$ and $n_t$ are large, both of order $O(\xi)$, where $\xi$ is a large number. Furthermore, we assume that $|\boldsymbol{\omega}_0 - \mathbf{w}_0|$ is of order $O(\xi^{-1/4})$. Using Lemma 2, together with the expressions of $B$ (Eq.68) and $\Gamma$ (Eqs.72,73), we can prove that

$$\frac{1}{p} BB^T = \left[ (1 - \alpha_t)^2 + \alpha_t^2 \frac{p+1}{n_t} \right] I_{n_v m} + O\left( \xi^{-1/2} \right) \tag{75}$$

$$\Gamma = \left\{ \nu^2 \left[ (1 - \alpha_t)^2 + \alpha_t^2 \frac{p+1}{n_t} \right] + \sigma^2 \left( 1 + \frac{\alpha_t^2 p}{n_t} \right) \right\} I_{n_v m} + O\left( \xi^{-1/2} \right) \tag{76}$$

$$B(\boldsymbol{\omega}_0 - \mathbf{w}_0)(\boldsymbol{\omega}_0 - \mathbf{w}_0)^T B^T = |\boldsymbol{\omega}_0 - \mathbf{w}_0|^2 \left[ (1 - \alpha_t)^2 + \alpha_t^2 \frac{p+1}{n_t} \right] I_{n_v m} + O\left( \xi^{-1/2} \right) \tag{77}$$

Using Eq.75 and Taylor expansion, the inverse $(BB^T)^{-1}$ is equal to

$$(BB^T)^{-1} = \frac{1}{p} \left[ (1 - \alpha_t)^2 + \alpha_t^2 \frac{p+1}{n_t} \right]^{-1} I_{n_v m} + O\left( \xi^{-3/2} \right), \tag{78}$$

Substituting the three expressions above in Eq.74, and ignoring terms of lower order, we find

$$\mathbb{E}\,|\boldsymbol{\omega}^\star - \mathbf{w}_0|^2 = \left( 1 - \frac{n_v m}{p} \right) |\boldsymbol{\omega}_0 - \mathbf{w}_0|^2 + \frac{n_v m}{p} \left[ \nu^2 + \sigma^2 \frac{1 + \frac{\alpha_t^2 p}{n_t}}{(1 - \alpha_t)^2 + \alpha_t^2 \frac{p+1}{n_t}} \right] + O\left( \xi^{-3/2} \right) \tag{79}$$

Substituting this expression into in Eq.65, we find the value of average test loss

$$\overline{\mathcal{L}}^{test} = \frac{\sigma^2}{2} \left( 1 + \frac{\alpha_r^2 p}{n_r} \right) + \tag{80}$$

$$+ h^r \left[ \frac{\nu^2}{2} \left( 1 + \frac{n_v m}{p} \right) + \frac{1}{2} \left( 1 - \frac{n_v m}{p} \right) |\boldsymbol{\omega}_0 - \mathbf{w}_0|^2 + \frac{\sigma^2 n_v m}{2p} \frac{1 + \frac{\alpha_t^2 p}{n_t}}{h^t} \right] + O\left( \xi^{-3/2} \right) \tag{81}$$

where we define the following expressions

$$h^t = (1 - \alpha_t)^2 + \alpha_t^2 \frac{p+1}{n_t} \quad \text{and} \quad h^r = (1 - \alpha_r)^2 + \alpha_r^2 \frac{p+1}{n_r} \tag{82}$$

### 7.3.2 Underparameterized case (Theorem 2)

In the underparameterized case ($p < n_v m$), under the assumption that the inverse of $B^T B$ exists, the value of $\boldsymbol{\omega}$ that minimizes Eq.66 is equal to

$$\boldsymbol{\omega}^\star = (B^T B)^{-1} B^T \boldsymbol{\gamma} \tag{83}$$

Note that the matrix $B$ does not depend on $\mathbf{w}, \mathbf{z}^t, \mathbf{z}^v$, and $\mathbb{E}_{\mathbf{w}} \mathbb{E}_{\mathbf{z}^t} \mathbb{E}_{\mathbf{z}^v} \gamma = B\mathbf{w}_0$. We denote by $\delta\gamma$ the deviation from the average, and we have

$$|\boldsymbol{\omega}^\star - \mathbf{w}_0|^2 = \mathrm{Tr}\left[\left(B^T B\right)^{-1} B^T \delta\gamma \, \delta\gamma^T B \left(B^T B\right)^{-1}\right] \tag{84}$$

We need to average this expression in order to calculate $\mathbb{E}|\boldsymbol{\omega}^\star - \mathbf{w}_0|^2$ in Eq.65. We start by averaging $\delta\gamma \, \delta\gamma^T$ over $\mathbf{w}, \mathbf{z}^t, \mathbf{z}^v$, since $B$ does not depend on those variables. Note that $\mathbf{w}, \mathbf{z}^t, \mathbf{z}^v$ are independent on each other and across tasks. As in previous section, we denote by $\Gamma$ the result of this operation, given by Eq.s72, 73. Finally, we need to average over the training and validation data

$$\mathbb{E}|\boldsymbol{\omega}^\star - \mathbf{w}_0|^2 = \mathop{\mathbb{E}}_{X^t} \mathop{\mathbb{E}}_{X^v} \mathrm{Tr}\left[\left(B^T B\right)^{-1} B^T \Gamma B \left(B^T B\right)^{-1}\right] \tag{85}$$

It is hard to average this expression because it includes nonlinear functions of the data. However, we can approximate these terms by assuming that either $m$ or $\xi$ (or both) is a large number, where $\xi$ is defined by assuming that both $n_t$ and $n_v$ are of order $O(\xi)$. Using Lemma 3, together with the expression of $B$ (Eq.68), and noting that each factor in Eq.85 has a sum over $m$ independent terms, we can prove that

$$\frac{1}{n_v m} B^T B = \left(1 - 2\alpha_t + \alpha_t^2 \mu_2\right) I_p + O\left((m\xi)^{-1/2}\right) \tag{86}$$

The expression for $\mu_2$ is given in Eq.32. Using this result and a Taylor expansion, the inverse is equal to

$$n_v m \left(B^T B\right)^{-1} = \left(1 - 2\alpha_t + \alpha_t^2 \mu_2\right)^{-1} I_p + O\left((m\xi)^{-1/2}\right) \tag{87}$$

Similarly, the term $B^T \Gamma B$ is equal to its average plus a term of smaller order

$$\frac{1}{n_v m} B^T \Gamma B = \frac{1}{n_v m} \mathbb{E}\left(B^T \Gamma B\right) + O\left((m\xi)^{-1/2}\right) \tag{88}$$

We substitute these expressions in Eq.85 and neglect lower orders. Here we show how to calculate explicitly the expectation of $B^T \Gamma B$. For ease of notation, we define the matrix $A^{t(i)} = I - \frac{\alpha_t}{n_t} X^{t(i)^T} X^{t(i)}$. Using the expressions of $B$ (Eq.68) and $\Gamma$ (Eqs.72,73), the expression for $B^T \Gamma B$ is given by

$$B^T \Gamma B = \sigma^2 \sum_{i=1}^m A^{t(i)^T} X^{v(i)^T} X^{v(i)} A^{t(i)} + \frac{\nu^2}{p} \sum_{i=1}^m \left(A^{t(i)^T} X^{v(i)^T} X^{v(i)} A^{t(i)}\right)^2 +$$

$$+ \frac{\alpha_t^2 \sigma^2}{n_t^2} \sum_{i=1}^m A^{t(i)^T} X^{v(i)^T} X^{v(i)} X^{t(i)^T} X^{t(i)} X^{v(i)^T} X^{v(i)} A^{t(i)} \tag{89}$$

We use Eqs.31, 32 to calculate the average of the first term in Eq.89

$$\mathop{\mathbb{E}}_{X^t} \mathop{\mathbb{E}}_{X^v} \sum_{i=1}^m A^{t(i)^T} X^{v(i)^T} X^{v(i)} A^{t(i)} = n_v m \left(1 - 2\alpha_t + \alpha_t^2 \mu_2\right) I_p \tag{90}$$

We use Eqs.31, 32, 33, 41, 36, 37, 38, 39 to calculate the average of the second term

$$\mathop{\mathbb{E}}_{X^t} \mathop{\mathbb{E}}_{X^v} \sum_{i=1}^m \left(A^{t(i)^T} X^{v(i)^T} X^{v(i)} A^{t(i)}\right)^2 = \mathop{\mathbb{E}}_{X^t} \sum_{i=1}^m \left[n_v \left(n_v + 1\right) A^{t(i)^4} + n_v A^{t(i)^2} \mathrm{Tr}\left(A^{t(i)^2}\right)\right] = \tag{91}$$

$$= m n_v \left(n_v + 1\right) \left(1 - 4\alpha_t + 6\alpha_t^2 \mu_2 - 4\alpha_t^3 \mu_3 + \alpha_t^4 \mu^4\right) I_p +$$
$$+ m n_v p \left(1 - 4\alpha_t + 2\alpha_t^2 \mu_2 + 4\alpha_t^2 \mu_{1,1} - 4\alpha_t^3 \mu_{2,1} + \alpha_t^4 \mu_{2,2}\right) I_p \tag{92}$$

Finally, we compute the average of the third term, using Eqs.31, 32, 33, 34, 41, 36, 37

$$\mathop{\mathbb{E}}_{X^t} \mathop{\mathbb{E}}_{X^v} \sum_{i=1}^m A^{t(i)^T} X^{v(i)^T} X^{v(i)} X^{t(i)^T} X^{t(i)} X^{v(i)^T} X^{v(i)} A^{t(i)} = \tag{93}$$

$$= \mathop{\mathbb{E}}_{X^t} \sum_{i=1}^m \left[n_v \left(n_v + 1\right) A^{t(i)^T} X^{t(i)^T} X^{t(i)} A^{t(i)} + n_v A^{t(i)^T} A^{t(i)} \mathrm{Tr}\left(X^{t(i)^T} X^{t(i)}\right)\right] = \tag{94}$$

$$= m n_v \left(n_v + 1\right) n_t \left(1 - 2\alpha_t \mu_2 + \alpha_t^2 \mu_3\right) I_p + m n_v n_t p \left(1 - 2\alpha_t \mu_{1,1} + \alpha_t^2 \mu_{2,1}\right) I_p \tag{95}$$

Putting everything together in Eq.85, and applying the trace operator, we find the following expression for the meta-parameter variance

$$\mathbb{E}\left|\boldsymbol{\omega}^{\star}-\mathbf{w}_0\right|^2 = \frac{p}{n_v m}\left(1-2\alpha_t+\alpha_t^2\mu_2\right)^{-2}\left\{\sigma^2\left(1-2\alpha_t+\alpha_t^2\mu_2\right)+\right.$$

$$+\frac{\alpha_t^2\sigma^2}{n_t}\left[(n_v+1)\left(1-2\alpha_t\mu_2+\alpha_t^2\mu_3\right)+p\left(1-2\alpha_t\mu_{1,1}+\alpha_t^2\mu_{2,1}\right)\right]$$

$$+\frac{\nu^2}{p}\left[(n_v+1)\left(1-4\alpha_t+6\alpha_t^2\mu_2-4\alpha_t^3\mu_3+\alpha_t^4\mu^4\right)+\right.$$

$$\left.\left.+p\left(1-4\alpha_t+2\alpha_t^2\mu_2+4\alpha_t^2\mu_{1,1}-4\alpha_t^3\mu_{2,1}+\alpha_t^4\mu_{2,2}\right)\right]\right\}+O\left((m\xi)^{-3/2}\right) \qquad (96)$$

We rewrite this expression as

$$\mathbb{E}\left|\boldsymbol{\omega}^{\star}-\mathbf{w}_0\right|^2 = \frac{p}{h^{t2}n_v m}\left\{\sigma^2\left[h^t+\frac{\alpha_t^2}{n_t}\left[(n_v+1)g_1+pg_2\right]\right]+\frac{\nu^2}{p}\left[(n_v+1)g_3+pg_3\right]\right\}+$$

$$+O\left((m\xi)^{-3/2}\right) \qquad (97)$$

where we defined the following expressions for $g_i$

$$g_1 = 1-2\alpha_t\mu_2+\alpha_t^2\mu_3 \qquad (98)$$
$$g_2 = 1-2\alpha_t\mu_{1,1}+\alpha_t^2\mu_{2,1} \qquad (99)$$
$$g_3 = 1-4\alpha_t+6\alpha_t^2\mu_2-4\alpha_t^3\mu_3+\alpha_t^4\mu^4 \qquad (100)$$
$$g_4 = 1-4\alpha_t+2\alpha_t^2\mu_2+4\alpha_t^2\mu_{1,1}-4\alpha_t^3\mu_{2,1}+\alpha_t^4\mu_{2,2} \qquad (101)$$

and $\mu_i$ are equal to

$$\mu_2 = \frac{1}{n_t}\left(n_t+p+1\right) \qquad (102)$$

$$\mu_3 = \frac{1}{n_t^2}\left(n_t^2+p^2+3n_tp+3n_t+3p+4\right) \qquad (103)$$

$$\mu_4 = \frac{1}{n_t^3}\left(n_t^3+p^3+6n_t^2p+6n_tp^2+6n_t^2+6p^2+17n_tp+21n_t+21p+20\right) \qquad (104)$$

$$\mu_{1,1} = \frac{1}{n_t^2p}\left(n_t^2p+2n_t\right) \qquad (105)$$

$$\mu_{2,1} = \frac{1}{n_t^2p}\left(n_t^2p+n_tp^2+n_tp+4n_t+4p+4\right) \qquad (106)$$

$$\mu_{2,2} = \frac{1}{n_t^3p}\left(n_t^3p+n_tp^3+2n_t^2p^2+2n_t^2p+2n_tp^2+8n_t^2+8p^2+21n_tp+20n_t+20p+20\right) \qquad (107)$$

Substituting this expression back into Eq.65 returns the final expression for the average test loss, equal to

$$\overline{\mathcal{L}}^{test} = \frac{\sigma^2}{2}\left(1+\frac{\alpha_r^2p}{n_r}\right)+\frac{h^r\nu^2}{2}+$$

$$+\frac{h^r}{2h^{t2}}\frac{p}{n_v m}\left\{\sigma^2\left[h^t+\frac{\alpha_t^2}{n_t}\left[(n_v+1)g_1+pg_2\right]\right]+\frac{\nu^2}{p}\left[(n_v+1)g_3+pg_4\right]\right\}+O\left((m\xi)^{-3/2}\right) \qquad (108)$$

## 7.4 PROOF OF THEOREM 3

In this section, we release some assumption on the distributions of data and parameters. In particular, we do not assume a specific distribution for input data vectors $\mathbf{x}$ and generating parameter vector

$\mathbf{w}$, besides that different data vectors are independent, and so are data and parameters for different tasks. We further assume that those vectors have zero mean, and denote their covariance as

$$\Sigma = \mathbb{E}\mathbf{x}\mathbf{x}^T \tag{109}$$

$$\Sigma_w = \mathbb{E}\mathbf{w}\mathbf{w}^T \tag{110}$$

We will also use the following matrix, including fourth order moments

$$F = \mathbb{E}\left(\mathbf{x}^T\Sigma\mathbf{x}\right)\mathbf{x}\mathbf{x}^T \tag{111}$$

We do not make any assumption about the distribution of $\mathbf{x}$, but we note that, if $\mathbf{x}$ is Gaussian, then $F = 2\Sigma^3 + \Sigma\text{Tr}\left(\Sigma^2\right)$. We keep the assumption that the output noise is Gaussian and independent for different data points and tasks, with variance $\sigma^2$. Using the same notation as in previous sections, we will also use the following expressions (for any $p \times p$ matrix $A$)

$$\mathbb{E}\left[X^T X\right] = n\Sigma \tag{112}$$

$$\mathbb{E}\,\text{Tr}\left[\Sigma X^T X A X^T X\right] = \text{Tr}\left\{A\left[n^2\Sigma^3 + n\left(F - \Sigma^3\right)\right]\right\} \tag{113}$$

We proceed to derive the same formula under these less restrictive assumptions, in the overparameterized case only, following is the same derivation of section 7.3. We further assume $\boldsymbol{\omega}_0 = 0$, $\mathbf{w}_0 = 0$. Again we start from the expression in Eq.24 for the test output, and we rewrite the test loss in Eq.27 as

$$\overline{\mathcal{L}}^{test} = \mathbb{E}\frac{1}{2n_s}\left|X^s\left(\mathbf{w}' - \boldsymbol{\theta}^\star\right) + \mathbf{z}^s\right|^2 \tag{114}$$

We average this expression with respect to $X^s, \mathbf{z}^s$, noting that $\boldsymbol{\theta}^\star$ does not depend on test data. We further average with respect to $\mathbf{w}'$, but note that $\boldsymbol{\theta}^\star$ depends on test parameters, so we average only terms that do not depend on $\boldsymbol{\theta}^\star$. Using Eq.112, the result is

$$\overline{\mathcal{L}}^{test} = \frac{\sigma^2}{2} + \frac{1}{2}\text{Tr}\left(\Sigma\Sigma_w\right) + \mathbb{E}\left[\frac{1}{2}\boldsymbol{\theta}^{\star T}\Sigma\,\boldsymbol{\theta}^\star - \mathbf{w}'^T\Sigma\,\boldsymbol{\theta}^\star\right] \tag{115}$$

The second term in the expectation is linear in $\boldsymbol{\theta}^\star$ and can be averaged over $X^r, \mathbf{z}^r$, using Eq.25 and noting that $\boldsymbol{\omega}^\star$ does not depend on target data. The result is

$$\mathbb{E}_{X^r}\mathbb{E}_{\mathbf{z}^r}\boldsymbol{\theta}^\star = (I - \alpha_r\Sigma)\boldsymbol{\omega}^\star + \alpha_r\Sigma\mathbf{w}' \tag{116}$$

Furthermore, we show below (Eq.128) that the following average holds

$$\mathbb{E}_{\mathbf{w}}\mathbb{E}_{\mathbf{z}^t}\mathbb{E}_{\mathbf{z}^v}\boldsymbol{\omega}^\star = 0 \tag{117}$$

Combining Eqs.116, 117, we can calculate the second term in the expectation of Eq.115 and find

$$\overline{\mathcal{L}}^{test} = \frac{\sigma^2}{2} + \frac{1}{2}\text{Tr}\left(\Sigma\Sigma_w\right) - \alpha_r\text{Tr}\left(\Sigma^2\Sigma_w\right) + \mathbb{E}\frac{1}{2}\boldsymbol{\theta}^{\star T}\Sigma\,\boldsymbol{\theta}^\star \tag{118}$$

We start by averaging the third term of this expression over $\mathbf{z}^r, \mathbf{w}'$, using Eq.25 and noting that $\boldsymbol{\omega}^\star$ does not depend on target data and test parameters. The result is

$$\mathbb{E}_{\mathbf{w}'}\mathbb{E}_{\mathbf{z}^r}\boldsymbol{\theta}^{\star T}\Sigma\,\boldsymbol{\theta}^\star = \text{Tr}\left[\Sigma\left(I - \frac{\alpha_r}{n_r}X^{rT}X^r\right)\boldsymbol{\omega}^\star\boldsymbol{\omega}^{\star T}\left(I - \frac{\alpha_r}{n_r}X^{rT}X^r\right)\right] + \tag{119}$$

$$+ \frac{\alpha_r^2\sigma^2}{n_r^2}\text{Tr}\left[X^r\Sigma X^{rT}\right] + \frac{\alpha_r^2}{n_r^2}\text{Tr}\left[\Sigma X^{rT}X^r\Sigma_w X^{rT}X^r\right] \tag{120}$$

We now average over $X^r$, again noting that $\boldsymbol{\omega}^\star$ does not depend on target data. Using Eqs.112, 113, we find

$$\mathbb{E}_{X^r}\mathbb{E}_{\mathbf{w}'}\mathbb{E}_{\mathbf{z}^r}\boldsymbol{\theta}^{\star T}\Sigma\,\boldsymbol{\theta}^\star = \text{Tr}\left\{\boldsymbol{\omega}^\star\boldsymbol{\omega}^{\star T}\left[\Sigma\left(I - \alpha_r\Sigma\right)^2 + \frac{\alpha_r^2}{n_r}\left(F - \Sigma^3\right)\right]\right\} + \tag{121}$$

$$+ \frac{\alpha_r^2\sigma^2}{n_r}\text{Tr}\left(\Sigma^2\right) + \alpha_r^2\text{Tr}\left\{\Sigma_w\left[\Sigma^3 + \frac{1}{n_r}\left(F - \Sigma^3\right)\right]\right\} \tag{122}$$

We can now rewrite the average test loss in Eq.118 as

$$\overline{\mathcal{L}}^{test} = \frac{\sigma^2}{2}\left[1 + \frac{\alpha_r^2}{n_r}\text{Tr}\left(\Sigma^2\right)\right] + \frac{1}{2}\text{Tr}\left[\left(\Sigma_w + \mathbb{E}\,\boldsymbol{\omega}^\star\boldsymbol{\omega}^{\star T}\right)H^r\right] \tag{123}$$

where we define the following matrix

$$H^r = \left[\Sigma\left(I - \alpha_r\Sigma\right)^2 + \frac{\alpha_r^2}{n_r}\left(F - \Sigma^3\right)\right] \tag{124}$$

In order to average the last term, we need an expression for $\boldsymbol{\omega}^\star$. We note that the loss in Eq.20 is quadratic in $\boldsymbol{\omega}$, therefore the solution in Eq.22 can be found using standard linear algebra. In particular, the loss in Eq.20 can be rewritten as

$$\mathcal{L}^{meta} = \frac{1}{2n_v m}\left|\boldsymbol{\gamma} - B\boldsymbol{\omega}\right|^2 \tag{125}$$

where $\boldsymbol{\gamma}$ is a vector of shape $n_v m \times 1$, and $B$ is a matrix of shape $n_v m \times p$. The vector $\boldsymbol{\gamma}$ is a stack of $m$ vectors

$$\boldsymbol{\gamma} = \begin{pmatrix} X^{v(1)}\left(I - \frac{\alpha_t}{n_t}X^{t(1)T}X^{t(1)}\right)\mathbf{w}^{(1)} - \frac{\alpha_t}{n_t}X^{v(1)}X^{t(1)T}\mathbf{z}^{t(1)} + \mathbf{z}^{v(1)} \\ \vdots \\ X^{v(m)}\left(I - \frac{\alpha_t}{n_t}X^{t(m)T}X^{t(m)}\right)\mathbf{w}^{(m)} - \frac{\alpha_t}{n_t}X^{v(m)}X^{t(m)T}\mathbf{z}^{t(m)} + \mathbf{z}^{v(m)} \end{pmatrix} \tag{126}$$

Similarly, the matrix $B$ is a stack of $m$ matrices

$$B = \begin{pmatrix} X^{v(1)}\left(I - \frac{\alpha_t}{n_t}X^{t(1)T}X^{t(1)}\right) \\ \vdots \\ X^{v(m)}\left(I - \frac{\alpha_t}{n_t}X^{t(m)T}X^{t(m)}\right) \end{pmatrix} \tag{127}$$

In the overparameterized case ($p > n_v m$), under the assumption that the inverse of $BB^T$ exists, the value of $\boldsymbol{\omega}$ that minimizes Eq.125, and that also has minimum norm, is equal to

$$\boldsymbol{\omega}^\star = B^T\left(BB^T\right)^{-1}\boldsymbol{\gamma} \tag{128}$$

Note that the matrix $B$ does not depend on $\mathbf{w}, \mathbf{z}^t, \mathbf{z}^v$, and $\mathbb{E}_{\mathbf{w}}\,\mathbb{E}_{\mathbf{z}^t}\,\mathbb{E}_{\mathbf{z}^v}\;\boldsymbol{\gamma} = 0$, therefore Eq.117 holds. In order to finish calculating Eq.123, we need to average the following term

$$\text{Tr}\left(H^r\boldsymbol{\omega}^\star\boldsymbol{\omega}^{\star T}\right) = \text{Tr}\left[\left(BB^T\right)^{-1}\boldsymbol{\gamma}\boldsymbol{\gamma}^T\left(BB^T\right)^{-1}\left(BH^rB^T\right)\right] \tag{129}$$

where we used the cyclic property of the trace. We start by averaging $\boldsymbol{\gamma}\boldsymbol{\gamma}^T$ over $\mathbf{w}, \mathbf{z}^t, \mathbf{z}^v$, since $B$ does not depend on those variables. Note that $\mathbf{w}, \mathbf{z}^t, \mathbf{z}^v$ are independent on each other and across tasks. We denote by $\Gamma$ the result of this operation, which is equal to a block diagonal matrix

$$\Gamma = \mathbb{E}_{\mathbf{w}}\,\mathbb{E}_{\mathbf{z}^t}\,\mathbb{E}_{\mathbf{z}^v}\,\boldsymbol{\gamma}\boldsymbol{\gamma}^T = \begin{pmatrix} \Gamma^{(1)} & 0 & 0 \\ 0 & \ddots & 0 \\ 0 & 0 & \Gamma^{(m)} \end{pmatrix} \tag{130}$$

Where matrix blocks are given by the following expression

$$\Gamma^{(i)} = X^{v(i)}\left(I - \frac{\alpha_t}{n_t}X^{t(i)T}X^{t(i)}\right)\Sigma_w\left(I - \frac{\alpha_t}{n_t}X^{t(i)T}X^{t(i)}\right)X^{v(i)T} + \tag{131}$$

$$+ \sigma^2\left(I_{n_v} + \frac{\alpha_t^2}{n_t^2}X^{v(i)}X^{t(i)T}X^{t(i)}X^{v(i)T}\right) \tag{132}$$

Finally, we need to average over the training and validation data

$$\mathbb{E}\,\text{Tr}\left(H^r\boldsymbol{\omega}^\star\boldsymbol{\omega}^{\star T}\right) = \mathbb{E}_{X^t}\,\mathbb{E}_{X^v}\,\text{Tr}\left[\left(BB^T\right)^{-1}\Gamma\left(BB^T\right)^{-1}\left(BH^rB^T\right)\right] \tag{133}$$

These averages are hard to compute since they involve nonlinear functions of the data. However, we can approximate these terms by assuming that $p$ and $n_t$ are large, both of order $O(\xi)$, where $\xi$ is a large number. Furthermore, we assume that $\text{Tr}\left(\Sigma_w^2\right)$ is of order $O\left(\xi^{-1}\right)$, and that the variances of matrix products of the rescaled inputs $\mathbf{x}/\sqrt{p}$, up to sixth order, are all of order $O\left(\xi^{-1}\right)$, in particular

$$\text{Var}\left(\frac{1}{p}X^{v(i)}X^{v(j)^T}\right) = O\left(\xi^{-1}\right) \tag{134}$$

$$\text{Var}\left(\frac{1}{p^2}X^{v(i)}X^{t(i)^T}X^{t(i)}X^{v(j)^T}\right) = O\left(\xi^{-1}\right) \tag{135}$$

$$\text{Var}\left(\frac{1}{p^3}X^{v(i)}X^{t(i)^T}X^{t(i)}X^{t(j)^T}X^{t(j)}X^{v(j)^T}\right) = O\left(\xi^{-1}\right) \tag{136}$$

Then, using Eqs.112, 113 and the expressions of $B$ (Eq.127) and $\Gamma$ (Eqs.130,131), we can prove that

$$BB^T = \text{Tr}\left(H^t\right)I_{n_v m} + O\left(\xi^{1/2}\right) \tag{137}$$

$$\Gamma = \left\{\text{Tr}\left(\Sigma_w H^t\right) + \sigma^2\left[1 + \frac{\alpha_t^2}{n_t}\text{Tr}\left(\Sigma^2\right)\right]\right\}I_{n_v m} + O\left(\xi^{1/2}\right) \tag{138}$$

$$BH^r B^T = \text{Tr}\left(H^r H^t\right)I_{n_v m} + +O\left(\xi^{1/2}\right) \tag{139}$$

where, similar to Eq.124, we define

$$H^t = \left[\Sigma\left(I - \alpha_t\Sigma\right)^2 + \frac{\alpha_t^2}{n_t}\left(F - \Sigma^3\right)\right] \tag{140}$$

Note that all these terms are of order $O\left(\xi\right)$. The inverse of $BB^T$ can be found by a Taylor expansion

$$\left(BB^T\right)^{-1} = \text{Tr}\left(H^t\right)^{-1}I_{n_v m} + O\left(\xi^{-3/2}\right) \tag{141}$$

Substituting these expressions in Eq.133, we find

$$\mathbb{E}\,\text{Tr}\left(H^r\boldsymbol{\omega}^\star\boldsymbol{\omega}^{\star T}\right) = n_v m\frac{\text{Tr}\left(H^r H^t\right)\left\{\text{Tr}\left(\Sigma_w H^t\right) + \sigma^2\left[1 + \frac{\alpha_t^2}{n_t}\text{Tr}\left(\Sigma^2\right)\right]\right\}}{\text{Tr}\left(H^t\right)^2} + O\left(\xi^{-3/2}\right) \tag{142}$$

Substituting this expression into in Eq.123, we find the value of average test loss

$$\overline{\mathcal{L}}^{test} = \frac{1}{2}\text{Tr}\left(\Sigma_w H^r\right) + \frac{\sigma^2}{2}\left[1 + \frac{\alpha_r^2}{n_r}\text{Tr}\left(\Sigma^2\right)\right] + \tag{143}$$

$$+ \frac{1}{2}n_v m\frac{\text{Tr}\left(H^r H^t\right)\left\{\text{Tr}\left(\Sigma_w H^t\right) + \sigma^2\left[1 + \frac{\alpha_t^2}{n_t}\text{Tr}\left(\Sigma^2\right)\right]\right\}}{\text{Tr}\left(H^t\right)^2} + O\left(\xi^{-3/2}\right) \tag{144}$$

