# OpenReview forum: "Meta-learning with negative learning rates"
_ICLR.cc/2021/Conference — ICLR 2021 Poster_

### Official Review · AnonReviewer2 · 2020-10-21
**Important theoretical study of MAML with counterintuitive findings, sheds light on empirical observations**

**Rating:** 8
**Confidence:** 4

**Review:**

Summary:
This paper gives much needed attention to the theoretical underpinnings of modern meta-learning algorithms such as MAML; it introduces a novel formal argument, discovers surprising implications and follows through to show that such predictions hold experimentally, despite being counterintuitive.


Strong points:
- Novel analysis of a practically important aspect of MAML-like meta-learning algorithms: setting the learning rates for training and adaptation; surprising theoretical result (of negative learning rates being optimal in some cases during meta-training) is well evaluated in controlled conditions.
- Honest discussion of limitations and good intuition is provided for applicability of the work. This is not too hard to do, but so many papers don’t provide it. Great job!
- Writing is clear enough, although the paper is dense.
- The authors don’t discuss previous empirical works (e.g. Meta-SGD/LEO) where meta-learning of the learning rate leads to negative inner-loop learning rates for some parameters, but such experiments actually provide further evidence to back up their claims, this time in SOTA deep models.


Weak points:
- Unfortunately, the paper includes only toy-task experiments, even by the standards of  meta-learning research.


Recommendation and Rationale:
I strongly support acceptance because this paper contains much needed fundamental work on theoretical underpinnings of modern meta-learning.

---

> ### Author Response · Authors · 2020-11-18
> **Answer to Reviewer #2**
>
> We thank the reviewer for recognizing the value of our work.
> We will include the references to Meta-SGD and LEO in the revised version of the paper.
> We are also running additional experiments on non-linear regression, we hope that we will obtain some results before November 24th.

---

### Official Review · AnonReviewer1 · 2020-10-28

**Rating:** 6
**Confidence:** 3

**Review:**

### Summary
The authors of this paper prove that the optimal learning rate for MAML is negative under mixed linear regression and nonlinear regression with overparametrized models. They verify that theoretic bounds align with numerical experiments.

### Comments
* As the authors note, "theoretical work is still lagging behind," so theoretical research to explain advances in meta-learning is useful.
* I think the clarity could be improved, especially in section 3. It was a bit difficult to understand notation and the exact setting. For instance, I think it'd be preferable to use a different Greek letter for the learning rates, rather than $\alpha_t$ and $\alpha_r$.
* I performed a surface level check of the proofs, and the results look correct to me.
* It would be nice to discuss more the practical implications of the results or whether there still exists a divide between theory and practice.

### Recommendation / Justification
I vote that the paper is marginally above the acceptance threshold. I think the results presented are interesting, but it is not clear to me what the implications are. It would be nice if the authors explored whether the setting they analyze aligns with more commonly used meta-learning tasks.

### Questions
* In the last paragraph of Section 3, are you referring to the learning rates at meta-training and meta-testing time?

### Minor feedback
* not only the performance -> not only [does] the performance
* These results help clarifying -> These results help clarify
* In general, I think it is better to reference equations as Equation 13 rather than 13.

---

> ### Author Response · Authors · 2020-11-17
> **Answer to Reviewer #1**
>
> We thank the reviewer for recognizing that the "theoretical research to explain advances in meta-learning is useful".
> Here we answer the reviewer's comments and questions point by point and we highlight the changes to be made (revised version to be uploaded soon).
>
> Comments
> - Thank you, we agree that theoretical research is useful.
> - We made a few changes to the text to make it more clear. For example, we state explicitly which equations correspond to the outer and to the inner loop of meta-learning, we explain more clearly how the training and validation data are drawn, and we specify that we only consider one step of gradient descent in the inner loop. We have not changed notation as we are running out of symbols and we could not agree on which symbols to use.
> - We also believe that proofs are correct, thank you. In the revised version of the paper we will present all mathematical results stated as formal theorems, as requested by other reviewers.
> - We are running additional experiments on non-linear regression, we hope that we will obtain some results before November 24th.
>
> Questions
> - Yes. We realize that "training" and "testing" is confusing, so we changed the text by adding the "meta-" prefix to both.
>
> Minor Feedback
> - Done
> - Done
> - Done

---

> > ### Comment · AnonReviewer1 · 2020-11-23
> > **Comments on response**
> >
> > Thank you for the response. I have read through the other reviews and responses.
> >
> > > We made a few changes to the text to make it more clear. For example, we state explicitly which equations correspond to the outer and to the inner loop of meta-learning, we explain more clearly how the training and validation data are drawn, and we specify that we only consider one step of gradient descent in the inner loop. We have not changed notation as we are running out of symbols and we could not agree on which symbols to use.
> >
> > Thanks, the new version reads a bit more clearly.
> >
> > > We are running additional experiments on non-linear regression, we hope that we will obtain some results before November 24th.
> >
> > Great, I think the addition will strengthen the paper.
> >
> > AnonReviewer3 raises some valid, unresolved concerns, which I hope you will address before the end of the rebuttal period.

---

> > > ### Author Response · Authors · 2020-11-24
> > > **Revised version**
> > >
> > > We have now uploaded a new version of the paper. Going back to your comment #2, we improved the clarity of Section 3 by adding a figure (Figure 1) that shows the graphical model of data generation.
> > > Since you mentioned the concerns by AnonReviewer3, we have also made the following major changes:
> > > - The mathematical results are now stated as three formal theorems, in sections 5.1, 5.2, 5.3, that include the order of the approximation error and the required assumptions. We added three lemmas in the Appendix that are instrumental to compute such error (section 7.2).
> > > - We added an experiment of a two-layer feed-forward neural network applied to a nonlinear (quadratic) regression problem, in section 5.4, showing that a negative learning rate indeed works better in this simple problem.
> > > - We show additional experiments on additional parameter sets in Figure 6 in the Appendix, showing more evidence in favor of our theory.

---

### Official Review · AnonReviewer4 · 2020-10-31
**Interesting findings on negative learning rates in meta learning**

**Rating:** 6
**Confidence:** 4

**Review:**

Summary:

This paper uses random matrix theory to study meta-learning in mixed linear regression and finds that the optimal learning rate for the inner loop/training is negative in order to minimize the test/adaption error. The results are interesting and novel. However, there are some concerns regarding the practical relevance and presentation of the results.

Major comments:

1. Implementation of negative learning rates in practice: This paper provides an interesting perspective that a negative learning rate could reduce the test error. My concern is with a negative learning rate, the training loss $\mathcal{L}^{(i)}$ in Eq. (3)  increases and the algorithm may not converge (at least on the training sets). In practice, for example, on deep learning models motivated by this paper in the first paragraph, how do you decide when to stop training parameters $\theta^{(i)}$ and $\omega$? How could you use the results in this paper to provide some guidance?
2. Presentation of the main results: I would suggest the authors formally state the results in theorems or propositions. Currently,  the main results are presented in Eq. (5), (9), (10), (15), and (16), that seem to be informal and need clarification. First, how is $\bar{\mathcal{L}}^{test}$ defined? Second, what does $\simeq$ mean in this context? If it means ````" approximately equal to," then what is the order of the estimation error? Third, the results are derived using mean-squared loss as shown in Eq. (25) and (28) in the appendix. It is helpful to be explicit about the loss function in the main text. Fourth, the loss function does not seem to have a regularization term. In the overparameterized regime, would the model suffer from overfitting?
3. Experiments: Could you elaborate more on why the theory matches the experiment pretty well in Figures 1.a, 2.a, and 3.a, while not the case in Figures 1.b, 2.b, and 3.b (especially Figure 3.b)? If I understand correctly, the data generating process in the simulation follows the assumption in the main results. Is it because the estimation error in $\bar{\mathcal{L}}^{test}$ (the terms omitted in the RHS of $\simeq$) is not negligible in finite samples? Also, is the curve in Figures 1.b and 3.b robust to the choice of parameters? It may be helpful to include a few other simulation setups in the appendix.

---

> ### Author Response · Authors · 2020-11-17
> **Answer to Reviewer #4**
>
> We thank the reviewer for recognizing that the "results are interesting and novel".
> Here we answer the reviewer's comments point by point and we highlight the changes to be made (revised version to be uploaded soon).
> We hope that the reviewer will consider the quality of the paper to have increased.
>
> 1. The training loss in Eq.(2) does not increase, our procedure finds its unique minimum with respect to the meta-parameter $\omega$. The negative learning rate is in the inner loop, Eq.(3). Since only one step of gradient descent is taken in the inner loop, there is no problem of convergence.
>
> 2. In the revised version of the paper, before Novermber 24th, we will include all mathematical results stated as formal theorems. The average test loss is defined by Eq.32. We will provide the order of the estimation error in the revised paper. Overfitting in our model is expressed by the term $\left|\boldsymbol\omega_0-\mathbf{w}_0\right|^2$ in Eq.5.
>
> 3. The reviewer is correct, the error is due to higher orders. We will provide figures with results of theory/experiment on additional parameter sets in the appendix. Our theory holds in the asymptotic limit, either for extremely overparameterized or extremely underparameterized problems.

---

> ### Author Response · Authors · 2020-11-24
> **Revised version**
>
> We have now uploaded a new version of the paper that addresses your points 2 and 3:
> - The mathematical results are now stated as three formal theorems, in sections 5.1, 5.2, 5.3, that include the order of the approximation error and the required assumptions. We added three lemmas in the Appendix that are instrumental to compute such error (section 7.2).
> - We show additional experiments on additional parameter sets in Figure 6 in the Appendix, showing more evidence in favor of our theory.

---

### Official Review · AnonReviewer3 · 2020-11-06
**Potentially interesting but still incomplete investigation of step-sizes in meta-learning.**

**Rating:** 6
**Confidence:** 4

**Review:**

Summary:
This paper studies meta-learning in the mixed linear regression setting, focusing on the effect of the within-task step-size on performance. For over-parameterized, under-parameterized, and NTK regimes they derive expressions for test-time loss that suggest that negative or close-to-zero learning rates are optimal, and provide experiments that closely match these results. However, some aspects of the mathematical approach are unclear, and the work's impact is limited without an investigation of the consequences of the analysis.

Strengths:
1. Understanding meta-learning in simple settings and focusing on the effect of learning rate are worthwhile goals.
2. The authors provide experimental evidence closely following the outlined theoretical results.

Weaknesses:
1. The mathematical results in the paper are difficult to follow carefully. It is still unclear to me what objective is being solved by the meta-training procedure, and various derivations in the appendix seem non-rigorous, such as replacing denominator terms by expectations. Formal statements and some proof sketches would be helpful. See also Questions 1-4.
2. No experimental investigation beyond the limited settings studied theoretically. The authors’ investigation leads to a clear prescriptive suggestion—use of negative within-task step-sizes during meta-training—that should be investigated on actual data to get a sense of whether the analysis extends to practical settings. This would be useful to do in both linear settings (see e.g. the experiments in Denevi et al. (2019)) and for standard meta-learning tasks. See also Questions 5-6.
3. The analysis is limited in not being able to handle representation learning or the case of label shuffling among tasks. Motivating empirical work such as the paper of Raghu et al. (2020) suggests that representation learning is a key component of being able to meta-learn without inner loop updates, while theoretical work (e.g. Saunshi et al. (2020)) suggests that the linear setting studied here cannot account for successful meta representation learning.

Questions:
1. The paper seems to suggest the objective ignores task structure completely, but in this case how does the within-task step-size affect meta-training at all?
2. How many steps of MAML are used in the analysis?
3. The line “All of the above distributions apply independently to each task and dataset (training, validation, target, test).” seems to suggest that data for the same task is drawn from different distributions - is this the case and if so how it this at all justifiable in the meta-learning setting?
4. In what references can the Gaussian moment results (36-46) be found?
5. How closely do experiments follow the analysis in non-asymptotic settings?
6. In the NTK results, why not investigate agreement between theory and experiments using kernel matrices obtained from actual networks rather than probability distributions on symmetric matrices?

Notes:
1. “their performance doesn’t seem to stop improving when adding more data and computing resources” - why would we expect it to?
2. “A meta-learning problem is solved by a bi-level optimization procedure: an outer loop optimizes meta-parameters across tasks, while an inner loop optimizes parameters within each task (Hospedales et al. (2020)).” - not all algorithms do this, c.f. Reptile (Nichol et al., 2018).
3. “recent papers argue that a simple alternative to meta-learning is just good enough, in which the inner loop is removed entirely (Chen et al. (2020a), Tian et al. (2020), Dhillon et al. (2020), Chen et al. (2020b), Raghu et al. (2020)). This is surprising, because this approach merges all source tasks into a single big training set, and it does not even distinguish between different source tasks during training.” - the paper of Raghu et al. (2020) does not remove the inner loop during training and still distinguishes between source tasks, since gradient updates still use data from the same task. Actually merging all data into a single big training set was shown to perform poorly by Finn et al. (2017).
4. “In the problem of mixed linear regression, we prove that the optimal learning rate is always negative in overparameterized models. The same result holds in underparameterized models for small values of the learning rate.” - the second sentence is unclear. The learning rate is negative when it is small?
5. “none of these studies look at the effect of the learning rate” - in the convex setting, Khodak et al. (2019) connect the within-task learning rate to the task similarity.
6. “Note that each task is characterized by a different distribution of the data, and we separate the training Dt and validation data Dv.”  - it is unclear whether these distributions are empirical distributions or population distributions. If they are empirical distributions then minimizing (1) is not the “goal of meta-learning” but an objective used to achieve the goal. If they are population distributions, are Dt and Dv different? Why and how?
7. What does the equality-like symbol in (5) mean? “Asymptotically equal to”?
8. Numerous equation reference numbers are missing parentheses.
9. “learning of the meta-parameter ω is performed by the outer loop” - still unclear which objective is being used for this. (1)?

References:
- Denevi, Stamos, Ciliberto, Pontil. “Online-within-online meta-learning.” NeurIPS 2019.
- Khodak, Balcan, Talwalkar. “Adaptive gradient-based meta-learning methods”. NeurIPS 2019.
- Nichol, Achiam, Schulman. “On first-order meta-learning algorithms.” arXiv 2018.

# Update after rebuttal phase

Thank you to the authors for engaging with reviewer comments. I think the paper is much clearer now, and the additional results in Figures 5 and 6 indicate that the analysis may be relevant for practical meta-learning settings. I am not sure of the necessity of the new data-generating plot for mixed linear regression in Figure 1 (my uncertainty here was resolved with words); the authors might consider using the space for putting Figure 6 (currently in the appendix) in the main paper, or for additional experiments. Two more notes:
1. It may make the paper easier to read if the appendix were part of the same PDF as the main paper and not in the supplementary material.
2. While the experiments are perhaps not difficult to reproduce, code would be helpful to the community.

I am increasing my rating to a 6, as I believe the paper presents an interesting result with sufficient evidence. I am not giving a higher rating as I think the paper's impact would increase substantially with experiments on actual data, either in the mixed linear or deep net setting. For future versions of the paper, I encourage the authors to consider adding such results.

---

> ### Author Response · Authors · 2020-11-17
> **Answer to Reviewer #3**
>
> We thank the reviewer for recognizing that "meta-learning in simple settings is a worthwhile goal", and for giving us several suggestions to improve the paper.
> Here we answer the reviewer's comments point by point and we highlight the changes to be made (revised version to be uploaded soon).
> We hope that the reviewer will consider the quality of the paper to have increased.
>
> Weaknesses:
> 1. In the revised version of the paper, before Novermber 24th, we will include all mathematical results stated as formal theorems.
> 2. We are running additional experiments on non-linear regression, we hope that we will obtain some results before November 24th.
> 3. By "label shuffling", we believe that the reviewer is referring to classification problems, where the set of classes are different across tasks. However, in this work we look at a regression problem only. In the context of regression, the work of Saunshi et al. (2020) is particularly interesting as they consider two-layer neural networks, where the output of the first layer is interpreted as the good representation. However, they study a different problem that has other limitations (e.g. they have only two tasks), and we believe that our study is also relevant for the general problem of "representation learning", since the general problem of finding a good meta-parameter across training tasks means finding a good representation for the meta-learning problem.
>
> Questions:
> 1. We are not sure what the reviewer is referring to here. Our objective in Eq.1 is a standard meta-learning objective, see e.g. Eqs. 5 and 6 in Hospedales et al. (2020).
> 2. We use one step of MAML in all theory and experiment. We now state this explicitly in Section 3, before Eq.3: "In this work we consider the simple case of a single gradient step"
> 3. This is our mistake. We now say: "The generating parameter w is drawn once and kept fixed within a task, and drawn independently for different tasks."
> 4. We could not find a reference with Gaussian moments up to all orders, so we calculated them ourselves using Isserlis Theorem and combinatorics.
> 5. We will provide figures with results of theory/experiment on additional parameter sets in the appendix, showing examples in non-asymptotic setting.
> 6. This is a good point, but unfortunately we are not able to do it in one week and it will the focus of our future work.
>
> Notes:
> 1. One usually expects performance to improve with data and compute (except in pathological cases, see e.g. Nakkiran 2019), but the question is whether it improves significantly. Today everyone assumes that deep learning models keep improving significantly with data and compute, but this was not obvious 10 years ago.
> 2. While we do not argue that all meta-learning algorithms have the inner/outer structure, we refer to Hospedales et al. (2020) for a definition of the meta-learning problem (as stated in the text). Furthermore, we believe that Reptile has also the inner/outer structure: Algorithm 1 of the Reptile paper shows the outer loop, and the inner loop is hidden in line 4 of the algorithm ("Compute $\tilde{\phi}$ with $k$ steps of SGD or Adam.")
> 3. In the paper by Raghu et al. (2020), most of the experiments are on the ANIL model (Almost No Inner Loop, in which the inner loop runs only on the head of the network). However, some experiments are on the NIL model (No Inner Loop, in which the inner loop is removed entirely, by removing the head of the network during testing). They found that NIL performs as good or even better than ANIL.
> 4. Here we mean that, when the optimal learning rate is small in magnitude, then it has to be negative, because the average test loss has a positive derivative at zero (see Eq.11). We modified the text now to: "The same result holds in underparameterized models provided that the optimal learning rate is small in absolute value."
> 5. We added the reference of Khodak et al. (2020) to our "Related Work" section. However, note that they study the learning rate of the outer loop (which we believe should never be negative), while we focus on the inner loop learning rate only.
> 6. This is our mistake, Dt and Dv are population distribution. We now say: "Within each task, the training Dt and validation data Dv are drawn independently from the same distribution.
> 7. Yes. We acknowledge that this notation is not very clear, so we will use a formal notation in the revised version.
> 8. In the entire paper, we don't use parentheses to reference equations. We believe that this notation is not ambiguous, because all equations are hyper-referenced, and can be located by a simple click on the pdf file. However, we now refer to every equation by "Eq."
> 9. Yes, we added a reference to the equation to make it clear.
>
> References:
> Nakkiran (2019) "More Data Can Hurt for Linear Regression: Sample-wise Double Descent." https://arxiv.org/abs/1912.07242
> Hospedales et al (2020) "Meta-Learning in Neural Networks: A Survey." https://arxiv.org/abs/2004.05439

---

> > ### Comment · AnonReviewer3 · 2020-11-19
> > **Reviewer follow-up**
> >
> > Thank you for the detailed response; I look forward to reading the revision and updating my evaluation accordingly. Below I’ve followed up on a few of your responses to my original questions/comments.
> >
> > Weaknesses:
> > 3.1 - On the question of label shuffling in point 3, note that in the regression setting one can effectively have label shuffling by flipping signs of the target variables.
> > 3.2 - On the question of representation learning in point 3, I believe representation learning should be understood as a nontrivial representation of the data that is then passed to a linear predictor, whereas here there is only a linear predictor without any nontrivial representation. Note that I do not believe this issue to be a flaw of the work, just a limitation.
> >
> > Questions:
> > 1 - My question here referred to the motivating line “this approach merges all source tasks into a single big training set, and it does not even distinguish between different source tasks during training” from the introduction; it does not seem accurate to me that the cited approaches do not distinguish between source tasks, since as far as I know gradients are still obtained by sampling data points from the same task.
> > 4 - I think this is important detail that should be included and relevant theorems should be cited.
> >
> > Notes:
> > 2 - I agree that Reptile has an inner loop, but I am not sure it is accurate to say it uses bilevel optimization since there is no separation of within-task data into training and validation sets.
> > 3 - As also noted in Question 1 above, I think my main concern here is with the “does not even distinguish between different source tasks during training” statement, which I do not believe is accurate.
> > 5 - As I noted in my original comment, they connect the task similarity to the within-task learning rate, which is the inner loop learning rate you focus on. I believe the outer loop learning rate is discussed separately.
> > 8 - Note that some readers still read on paper, not PDF. Adding Eq. also works.

---

> > > ### Author Response · Authors · 2020-11-24
> > > **Revised version**
> > >
> > > Thank you for your additional suggestions to improve our paper.
> > > We have now uploaded a new version of the paper addressing both your major and your minor concerns.
> > >
> > > Major changes:
> > > - The mathematical results are now stated as three formal theorems, in sections 5.1, 5.2, 5.3, that include the order of the approximation error and the required assumptions. We added three lemmas in the Appendix that are instrumental to compute such error (section 7.2).
> > > - We added an experiment of a two-layer feed-forward neural network applied to a nonlinear (quadratic) regression problem, in section 5.4, showing that a negative learning rate indeed works better in this simple problem.
> > > - We show additional experiments on additional parameter sets in Figure 6 in the Appendix, showing more evidence in favor of our theory.
> > > - We added a figure (Figure 1 in section 3) to explain more clearly the data generating model.
> > >
> > > Additional minor changes:
> > > - We understand your point now and agree, so we removed the sentence: "This is surprising, because this approach merges all source tasks into a single big training set, and it does not even distinguish between different source tasks during training."
> > > - In the "Related Work" section, we added: "The theoretical work of Khodak et al. (2019) connects the learning rate to task similarity."

---

### Decision · Program_Chairs · 2021-01-07
**Final Decision**

**Decision:**

Accept (Poster)

**Comment:**

While this paper would be significantly improved with experiments on real data, the reviewers all agreed that there is value in the ideas and simple experiments in this paper and all voted for acceptance after the discussion period.

We encourage the authors to consider adding an experimental evaluation in more realistic settings (e.g. with real data) in the final version of the paper.